

# Fast response methods for aero-elastic floating wind turbine design

Bogdan Pamfil[1], Henrik Bredmose[1], Taeseong Kim[1], and Wei Yu[2]

[1]DTU Wind and Energy Systems, Koppels Allé, Building 403, DK-2800 Kgs. Lyngby, Denmark
[2]Delft University of Technology, Wind energy section, Faculty of Aerospace Engineering, Kluyverweg 1, 2629 HS Delft, Netherlands

**Correspondence:** Bogdan Pamfil (bopa@dtu.dk)

**Abstract.** Fast response calculations in the frequency domain are valuable during the initial design of floating wind turbines, where many design variants must be evaluated. A direct frequency-domain treatment of aeroelastic rotor loads is typically infeasible due to the azimuthal time dependence of the system matrices. To overcome this limitation, we introduce a perturbation-based formulation inspired by Hill's method, which reformulates the response equations into separate orders involving constant system matrices derived via Fourier decomposition. This enables accurate and efficient response computation using the Fast Fourier Transform (FFT). For comparison, a Laplace-based perturbation method is also developed using the Laplace transform instead of the Fourier transform. To evaluate the novel fast response methods, we develop an azimuthally periodic and fully linearized model of a floating wind turbine. The response to various load cases is computed under different inflow and floater motion conditions. The proposed Fourier-based fast response method achieves high accuracy, with peak and standard deviation errors of 2% and 3.5%, respectively, while reducing computation time to 2.5 s for a 4096 s simulation—significantly faster than linear (45 s) and time-domain (90 s) models. The single perturbation method offers an effective trade-off between accuracy and speed, making it suitable for design and optimization studies.

## 1 Introduction

During the concept development and optimization phase of floating wind turbine design, linearized models-based methods are useful to quickly calculate aerodynamic and hydrodynamic response from pre-computed rotor loads. Because a wide range of aerodynamic and hydrodynamic load cases are necessary to be tested, the development of a frequency-domain solver is a solution in providing computationally fast responses.

An example of a frequency-domain solver that leverages pre-computed rotor loads and damping is the QuLAF (Quick Load Analysis of Floating wind turbines) model. The QuLAF model was introduced by Pegalajar-Jurado et al. (2018) and its performance was further studied by Madsen et al. (2019). It was designed to solve the Equations of Motion (EOMs) in the frequency domain to increase the simulation speed and to facilitate the consideration of frequency-dependent effects, such as added hydrodynamic mass or radiation damping. Its efficiency relies on the Fast Fourier Transform (FFT) solution of a four Degrees of Freedom (DOFs) model that accounts for the floater surge, heave, and pitch and the first tower mode's modal amplitude. It considers the aerodynamic rotor loads to be concentrated as a thrust and moment located at the rotor hub height position and includes the surge, heave, and pitch forcing at the floater basis. QuLAF's features have been proven to be useful



notably for the geometric optimization of a TetraSpar floater (Pollini et al., 2023) for a 15 $MW$ wind turbine. Similarly, NREL's Response Amplitudes of Floating Turbines (RAFT) model (Hall et al., 2022) was developed to solve the dynamic response in the frequency domain using the EOMs (Hall et al., 2023), and has also been applied for control, optimization, and mooring analysis purposes (Hall et al., 2022; Zalkind and Bortolotti, 2024; Lozon et al., 2024). Another related fast solver is the

Simplified Low-Order Wind turbine (SLOW) model (Schlipf et al., 2013), which was initially developed for nonlinear model predictive control in floating wind energy applications, and has since been extended for control and optimization purposes (Lemmer et al., 2017, 2020a, b, 2021). Recent efforts from NTNU have also focused on applying frequency-domain solvers for floater and mooring designs (Abdelmoteleb and Bachynski-Polić, 2024, 2025).

These fast response solvers are beneficial in the preliminary design phase of floating wind turbines because FFT-based

linearized models substantially reduce computational cost compared to a time-domain model (TDM) or linear model (LM). However, current frequency-domain approaches are limited in accuracy—they cannot directly correct for nonlinear effects or capture the strong periodicity of rotor loads associated with blade rotation and higher-order harmonics.

To address this limitation, the present study introduces novel fast response methods for a LM that are executed with the FFT algorithm as in the QuLAF code. For comparison with Fourier-based methods, we implement a technique that uses the Laplace

transform to convert the solution from the $s$-domain to the time domain. The novelty of this study lies in the development of fast response methods based on the separation of time varying elements of the system matrices and corrections within a state-space formulation, which can be expressed in either the frequency or Laplace domain.

Building on this foundation, the present work extends frequency- and Laplace-domain methodologies to floating wind energy with a focus on blade aerodynamic loads rather than floater hydrodynamic and mooring loads. In our model, the novelty lies

in explicitly accounting for blade-dependent rotor loads in both the frequency- and Laplace-domain, rather than reducing them to thrust and moment at the rotor hub, as done in existing models such as QuLAF, RAFT, and SLOW. While previous studies modeled the EOM in state-space form without considering azimuthal blade load effects, our formulation includes these effects directly. Linearizing motion-coupled aerodynamic forces remains challenging because of the blades' passage through a disturbed inflow, leading to time-variant azimuthal dependencies in the system matrices. To enable the assembly of a constant

system transfer function, one needs to get around the azimuthal dependence of the system matrix.

The proposed fast response methods draw inspiration from Hill's method (Hill, 1886), which employs harmonic decomposition to convert a periodic system into a linear time-invariant one (LTI)—a process also related to Floquet's theory (Floquet, 1883). Our previous work (Pamfil et al., 2025) has shown that Hill's method produces stability analysis results consistent with Floquet's and Coleman's (Coleman et al., 1957) transformations, while avoiding the latter's limitations for two-bladed rotors

or cases with wind shear and gravity effects. Hill's method uses a truncated Fourier series to decompose the system matrix into a harmonic summation of constant terms, and the reliability and accuracy of this decomposition have been further validated also by Hansen (2016) and Skjoldan (2009). Consequently, we exploit this harmonic decomposition to enable fast response calculations based on the FFT algorithm (Cooley and Tukey, 1965) and to develop a Laplace-based method relying on the inverse of the Laplace transform.





The FFT algorithm (Cooley and Tukey, 1965) ensures computational efficiency in the frequency domain while the Laplace transform provides a pathway for response analysis through transfer functions represented in the $s$-domain. Although Laplace-domain computations require the inverse transformation to obtain time-domain results—often a computational bottleneck—various numerical inversion algorithms, such as Talbot's method (Talbot, 1979), Stehfest's algorithm (Stehfest, 1970), and the de Hoog continued fraction method (de Hoog et al., 1982), offer accurate solutions. In this study, MATLAB's symbolic inverse Laplace

function (`ilaplace`) is used for convenience.

The fast response methods in this paper are validated using a simplified four-DOF floating wind turbine model, incorporating floater pitch motion and blade flapwise deflection (first mode only), as well as dynamic stall and gravity effects. The rotor speed is assumed constant, and pitch control and tower deflection are neglected to reduce complexity. Multiple load cases are simulated, including steady, sheared, and turbulent inflow, and harmonic or stochastic floater pitch excitation. The accuracy for

the fast response calculation variants is analyzed for a given duration against benchmark results of the original LM. Metrics include the exceedance probability results and the Standard Deviation Relative Error (SDRE) with respect to the LM and CPU time. Results demonstrate that fast response methods achieve a good trade-off between speed and accuracy. With limited loss of accuracy, they can potentially accelerate the state-of-the-art TDMs such as HAWC2, OpenFAST, SIMA, and Bladed, making them promising tools for early-stage design and optimization of Floating Offshore Wind Turbines (FOWTs).

## 2    Model description

As illustrated in Fig. 1, the floating wind turbine model considered operates at a constant rotational speed $\Omega$ and has four structural DOFs, being the flapwise deflection of the three blades, denoted by $a_l$, and the floater's pitch angular motion, represented by $\xi_5$. Moreover, there is a hydrodynamic forcing moment $M_F$ exerted on the floater base which represents wave forcing. The dynamic rotor loads for each $l^{th}$ blade are applied in the rotor out-of-plane direction (along the $\hat{y}'$ axis), which

coincides with the blade deflection $u_l(r,t)$. These loads arise from a concentrated aerodynamic blade force $F_{l,aero}$ applied at a reference location $d$ from the hub, as well as from a constant velocity $V_0$ with the fluctuation $\Delta V_{0,l}$ generated by a sheared and turbulent inflow. The blade aerodynamic loads and the contributions to wind velocity from the constant velocity $V_0$, turbulent fluctuation $V_{0,l,\text{turb}}$, and sheared inflow $V_{0,l,\text{shear}}$ are illustrated in Fig. 1. More details about the considered floating wind turbine model has been elaborated by the author (Pamfil et al., 2025). In this paper, we expand the floating wind turbine model

with wind, wave and gravity loads for the blade elements of mass $m(r)$ and the hub and nacelle of cumulative mass $M$.

The floating wind turbine model (Pamfil et al., 2025) also includes Øye's dynamic stall model (Øye, 1991), through three additional aerodynamic DOFs. The blade deflection is approximated using a modal approach, where only the blade first flapwise mode $(1f)$ contribution is taken into account. The reason why the blade deflection approximation is computed using only the blade first flapwise mode is in order to alleviate the model's number of DOFs and thus simplify it. The structural properties

of the floating wind turbine, such as the blade first flapwise mode's characteristics (natural frequency $\omega_{1f}$, mode shape $\phi_{1f}$, and modal mass), the blade length $L_b$ and mass per unit length $m(r)$, the rotor hub height $H$ and the combined mass $M$ for the hub and nacelle, are all based on the DTU 10 MW reference wind turbine (Bak et al., 2013). The floater pitching moment



coefficient $K_{\xi_5}$, that describes the rotational stiffness (Pamfil et al., 2025), was chosen to achieve a floater pitch period of 28.57 s and thus a frequency of $f_{\xi_5} = 0.035$ Hz.

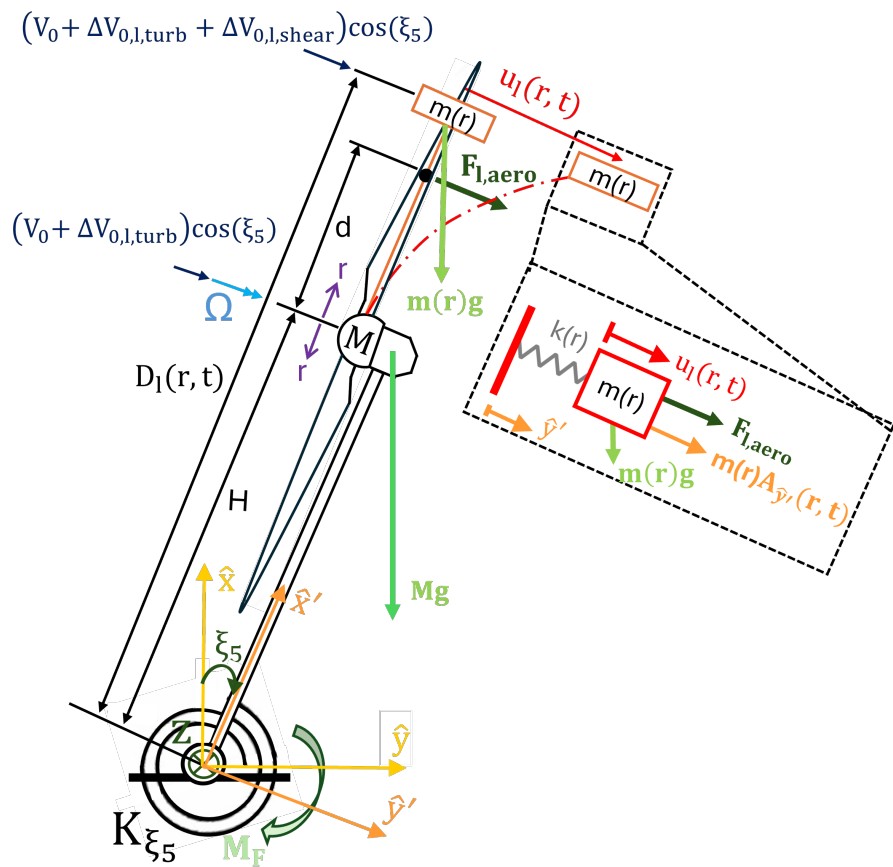

**Figure 1.** Floating wind turbine model where blade properties and parameters vary in the radial direction $r$ and with blade index $l$. This includes the blade deflection $u_l(r,t)$ and the distance of a blade element mass $m(r)$ from the floater base being $D_l(r,t)$.

The distance $d$ measured from the hub marks a reference radial position $r = d$ along the blade, at 70% of the blade's length ($d = 0.7L_b$), where the aerodynamic load on each blade, $F_{l,aero}$, is calculated. $F_{l,aero}$ is indicative of the load distribution along the entire blade.

Finally, it should be emphasized that this floating wind turbine model serves solely as a representative case study to illustrate the proposed fast response methods.

## 2.1 Equations of motion

For completeness, we provide a summary of the derivation of EOMs that we have already elaborated in previous work (Pamfil et al., 2025). To this end, we introduce the time-varying azimuthal angular position $\Psi_l(t)$ of a blade $l$, which is defined as $\Psi_l(t) = \frac{2\pi}{N_b}(l-1) + \Omega t$ and where $N_b = 3$ is the rotor's number of blades and $\Omega$ is its constant rotational speed.





For a non-deformed blade, the time-varying distance $D_l(r,t)$ of a blade element with mass $m(r)$ along the $\hat{x}'$ axis is calculated relative to the floater base position. It is also impacted by the radial position $r$ from the hub, yielding $D_l(r,t) = H + r\cos\Psi_l(t)$. For a deformed blade case, the local blade deflection $u_l(r,t)$ is computed as $u_l(r,t) = \phi_{1f}(r)a_l(t)$, where only the first flapwise (1f) mode shape $\phi_{1f}(r)$ with amplitude $a_l(t)$ is taken into consideration.

Subsequently, given a blade element's distance $D_l(r,t)$ from the floater basis and its deflection $u_l(r,t)$, its position $\hat{D}_l(r,t)$ is tracked as $\hat{D}_l(r,t) = D_l(r,t)\hat{x}'(t) + u_l(r,t)\hat{y}'(t)$, in the rotating coordinate system $(\hat{x}', \hat{y}')$. Fig. 1 shows that the unit vectors $\hat{x}'$ and $\hat{y}'$ can be expressed in terms of the global fixed coordinates $\hat{x}$ and $\hat{y}$ as $\hat{x}' = \cos(\xi_5)\hat{x} + \sin(\xi_5)\hat{y}$ and $\hat{y}' = -\sin(\xi_5)\hat{x} + \cos(\xi_5)\hat{y}$. This allows us to deduce the following time derivatives of $\hat{x}' = \dot{\xi}_5\hat{y}'$ and $\dot{\hat{y}}' = -\dot{\xi}_5\hat{x}'$. As previously demonstrated (Pamfil et al., 2025), these time derivatives are then used to lay out the kinematic equations for velocity $\hat{V}_l(r,t)$, acceleration $\hat{A}_l(r,t) = A_{\hat{x}',l}\hat{x}' + A_{\hat{y}',l}\hat{y}'$ and the rate of change of angular momentum $p_l(r,t)$ around the axis $\hat{z}' = \hat{x}' \times \hat{y}'$. The linearized rate of change of angular momentum, $p_{l,lin}(r,t)$, is detailed in Eq. (7) of our publication related to the current study (Pamfil et al., 2025).

Moving forward, the translational and rotational motion equations can be derived, along with the resulting EOM. To derive the Equation of Motion (EOM), we first establish the rotational motion equation for moments about the $\hat{z}$ axis as

$$\hat{z}: MH^2\ddot{\xi}_5\delta\xi_5 + K_{\xi_5}\xi_5\delta\xi_5 + MgH\sin\xi_5\delta\xi_5 +$$
$$\sum_{l=1}^{N_b}\left(\int_0^{L_b}(p_{l,lin}(r,t) + m(r)gD_l(r,t)\sin\xi_5)\,dr\,\delta\xi_5\right)$$
$$= M_F\delta\xi_5 + \underbrace{\left(\sum_{l=1}^{N_b}D_l(d,t)F_{l,aero}(t)\right)}_{M_{aero}}\delta\xi_5,$$

(1)

through the conservation of angular momentum for a virtual rotation $\delta\xi_5$. In Eq. (1), there is an internal angular momentum forcing component that considers the gravitational load contribution of the nacelle and hub having a cumulative mass $M$. The other gravitational angular momentum load contribution is accounted for by integrating the distributed mass $m(r)$ along the blade span with a moment arm $D_l(r,t)$. The external forcing arises from the applied floater pitch moment $M_F$ and the induced aerodynamic moment $M_{aero}$, which is generated by the aerodynamic loads $F_{l,aero}$ acting with the same moment arm $D_l(r,t)$.

In addition, the equation of translational motion along the $\hat{y}'$ axis for each blade is obtained as

$$\hat{y}: \int_0^{L_b}\left(\underbrace{m(r)A_{\hat{y}',l}(r,t)}_{f_{\hat{y}',l}(r,t)} + m(r)g\sin\xi_5\right)\delta a_l\phi_{1f}(r)\,dr +$$
$$\underbrace{\int_0^{L_b}k(r)a_l(t)\phi_{1f}(r)(\delta a_l\phi_{1f}(r))\,dr}_{K_{a_l}a_l\delta a_l}$$

(2)

$$= \underbrace{F_{l,aero}(t)\phi_{1f}(d)}_{GF_{a_l}}\delta a_l,$$





where the blade displacement virtual work is given by $\delta u_l(r,t) = \delta a_l(t)\phi_{1f}(r)$. In Eq. (2), the inertial force $f_{\hat{y}',l}(r,t) = m(r)A_{\hat{y}',l}(r,t)$ for a mass element $m(r)$ is influenced by the linearized tangential acceleration $A_{\hat{y}',l}(r,t)$, where only linear terms are retained. Here, $k(r)$ represents the blade sectional stiffness, defined as $k(r) = m(r)\omega_{1f}^2$, which is derived from the first flapwise natural frequency $\omega_{1f}$, while $\phi_{1f}(d)$ represents the value of the first flapwise mode shape at $r = d$. The blade internal gravitational load is obtained by integrating the load component perpendicular to the rotor plane along the blade span. The external blade force considered in Eq. (2) for each $l^{\text{th}}$ blade is denoted as the generalized aerodynamic blade force, $GF_{a_l}$.

The forcing terms for both the rotational and translation motion equations are implemented within the TDM forcing vector $\boldsymbol{F}_T$. The TDM's dynamics is described by the EOM for the structural DOFs vector of $\boldsymbol{x} = [\xi_5, a_1, a_2, a_3]^T$,

$$\mathbf{M}_S\ddot{\boldsymbol{x}} + \mathbf{C}_S\dot{\boldsymbol{x}} + \mathbf{K}_S\boldsymbol{x} = \boldsymbol{F}_T. \tag{3}$$

Eq. (3) includes the structural (index $S$) mass $\mathbf{M}_S$, damping $\mathbf{C}_S$ and stiffness $\mathbf{K}_S$ matrices. The structural mass and damping matrices, as defined in Eqs. (11) and (13), are taken from the model developed in our earlier work on floating wind turbine stability analysis (Pamfil et al., 2025). However, the stiffness matrix now includes gravitational terms, rendering it as follows:

$$\mathbf{K}_S = \begin{bmatrix} K_{\xi_5} + MgH & 0 & 0 & 0 \\ 0 & 0 & 0 & 0 \\ 0 & 0 & 0 & 0 \\ 0 & 0 & 0 & 0 \end{bmatrix} + \int_0^{L_b} m(r)\mathbf{K}_{S,b}(r)dr$$

$$\mathbf{K}_{S,b} = \begin{bmatrix} \Sigma_{l=1}^3 g\,D_l(r,t) & -\ddot{D}_1(r,t)\phi_{1f}(r) & -\ddot{D}_2(r,t)\phi_{1f}(r) & -\ddot{D}_3(r,t)\phi_{1f}(r) \\ g\,\phi_{1f}(r) & \omega_{1f}^2\phi_{1f}^2(r) & 0 & 0 \\ g\,\phi_{1f}(r) & 0 & \omega_{1f}^2\phi_{1f}^2(r) & 0 \\ g\,\phi_{1f}(r) & 0 & 0 & \omega_{1f}^2\phi_{1f}^2(r) \end{bmatrix}. \tag{4}$$

The gravity terms in the structural matrix $\mathbf{K}_S$ consider the small floater tilt assumption, such that $\sin\xi_5 \approx \xi_5$. Inspired from the Rayleigh damping approach, we have chosen the structural damping matrix $\mathbf{C}_S$ to be expressed as $\mathbf{C}_S = \mu\mathbf{K}_S$. In this expression, only the diagonal elements of the structural stiffness matrix $\mathbf{K}_S$, that are not affected by gravitational effects, are utilized within $\mathbf{C}_S$.

### 2.2 Floater pitch moment excitation

Regarding the floater pitch moment of excitation, $M_F$, it can be either harmonic or stochastic. The harmonic floater pitch moment $M_F$ has an amplitude $A_M$ and an excitation frequency $\Omega_M$, $M_F = A_M\cos(\Omega_M t)$, to model a regular wave forcing effect that affects the sinusoidal floater motion. The numerical value of $A_M = 1.212 \cdot 10^7$ Nm is selected to represent a realistic amplitude of the floater moment excitation for a floating wind turbine with the DTU 10 MW size and characteristics. As for the excitation frequency of $\Omega_M = 0.15 \cdot 2\pi$ rad s$^{-1}$, it is also representative of a typical sea-state.

On the other hand, the stochastic hydrodynamic moment $M_F$ that is considered in this paper, is extracted from simulations carried out for a water depth of $h = 320$ m for a representative spar floater. The spar-buoy floater's main specifications are its





submerged underwater draft, $\hat{x}_{Bot} = -120$ m, and its diameter, $D_{Spar} = 11.2$ m, that relates to the cylinder cross-sectional area $A_{Spar}$. These dimensions are representative of a 10 MW spar design.

Calculating the hydrodynamic moment for a spar-buoy floater greatly simplifies calculations through the usage of the Morison equation. To do so, we calculate first the hydrodynamic force $F_{hydro}$ per unit length, and next integrate up to the still water
level at $\hat{x} = 0$ after multiplication with the moment arm $\hat{x}$,

$$
\begin{aligned}
\tau_{hydro} &= \int_{\hat{x}_{Bot}}^{0} \hat{x} F_{hydro} d\hat{x} \\
&= \int_{\hat{x}_{Bot}}^{0} \hat{x}\rho_{water} C_m A_{Spar}\dot{u}_0 d\hat{x} + \int_{\hat{x}_{Bot}}^{0} \hat{x}\rho_{water} A_{Spar}\dot{u}_{wave} d\hat{x} \\
&\quad + \int_{\hat{x}_{Bot}}^{0} \hat{x}\rho_{water} C_{D,water} D_{Spar} u_0 |u_0| d\hat{x}.
\end{aligned}
\tag{5}
$$

The hydrodynamic force is calculated with no spar motion consideration. In this case, there is only a wave velocity and acceleration, $u_0 = u_{wave}$ and $\dot{u}_0 = \dot{u}_{wave}$, that are perceived by the spar-buoy. In Eq. (5), $\rho_{water}$ is the water density, $C_{D,water}$ is the drag coefficient and the inertia term $C_m$ is defined as $C_m = 1 + C_a$, where $C_a$ is the added mass coefficient. The
stochasticity in Eq. (5) arises from $u_{wave}$'s spectral decomposition for a number $N$ of wave frequency $\omega_{wave,j}$ samples with a stochastic phase shift $\epsilon_j$ applied,

$$
u_{wave}(\hat{x},\hat{y}) = \sum_{j=1}^{N} A_{wave,j}\omega_{wave,j}\frac{\cosh(k_{wave,j}(\hat{x}+h))}{\sinh(k_{wave,j}h)}\cos(\omega_{wave,j}t + \epsilon_j).
\tag{6}
$$

In Eq. (6), $A_{wave,j}$ is the wave amplitude obtained from the JONSWAP spectrum with a prescribed significant wave height $H_s = 1.2$ m, a peak period $T_p = 10$ s and a peak enhancement factor of $\gamma = 3.3$. The prescribed $H_s$ and $T_p$ values are correlated
to the water depth $h$ and to the inflow velocity $V_0$ for the operational state of the wind turbine. Additionally, $k_{wave,j}$ is the wave number solved through the wave dispersion relation, i.e. $\omega_{wave,j}^2 = g k_{wave,j} \tanh(k_{wave,j}h)$.

### 2.3   Aerodynamic load

The loading vector, $\boldsymbol{F}_T$, that is acting on the structure is not only influenced by the floater pitch hydrodynamic moment, $M_F$, but also by the aerodynamic loads. The aerodynamic loads, $F_{l,aero}$, acting on the blades are directly influenced by the lift
force, $L_l$, and evaluated at the reference radial position $r = d$, $L_l = \frac{1}{2}\rho\{cC_{L,l}V_{rel,l}^2\}_{r=d}$. The lift force $L_l$ is a function of air density $\rho$, the local airfoil relative velocity $V_{rel,l}$, the airfoil chord length $c$, and the dynamic lift coefficient $C_{L,l}$. The dynamic lift coefficient $C_{L,l}$ is computed through the dependency on the dynamic stall separation function, $f_{s,l}$, which has been more thoroughly explained in the associated publication (Pamfil et al., 2025). This is elaborated in Section 3.1 and Appendix B.



## 2.4 Inflow velocity fluctuation

The inflow velocity consists of a constant value $V_0$ at hub height $H$ and a fluctuation that is caused by a spatially coherent turbulent inflow $\Delta V_{0,turb}$ and a shear periodic variation $\Delta V_{0,l,shear}$, refer to Fig. 1. The presence of turbulence generates a variability in the wind speed that affects the rotor stochastic aerodynamic forces. We consider a deterministic linear shear velocity model for wind (Hansen, 2015) at hub height ($\hat{x} = H$) to determine the shear inflow velocity variation $\Delta V_{0,l,shear}$ around that point given the shear exponent $\nu_{shear} = 0.2$.

On top of the velocity variation $\Delta V_{0,l,shear}$, a turbulent spatially coherent variation $\Delta V_{0,turb}$ is taken also into account in the inflow velocity $V_{0,l} = V_0(\hat{x} = H) + \Delta V_{0,l}$ that is perceived by the $l^{th}$ blade. This results in

$$
\begin{aligned}
V_{0,l} =& V_0(H) + \left. \frac{\partial \left( V_0(H) \left( \frac{\hat{x}}{H} \right)^{\nu_{shear}} \right)}{\partial \hat{x}} \right|_{\hat{x}=H} \Delta_{\hat{x},l}(d) + \Delta V_{0,turb} \\
=& V_0(H) + \underbrace{V_0(H) \left( \frac{\nu_{shear} \, d \cos \Psi_l}{H} \right)}_{\Delta V_{0,l,shear}} + \Delta V_{0,turb},
\end{aligned}
\tag{7}
$$

where the radial distance from the hub, $\Delta_{\hat{x},l}(r) = r \cos \Psi_l$ is taken at $r = d$. The inflow velocity component that affects the system dynamics is its projection in the normal direction to the rotor plane, $V_{0,l} \cos(\xi_5)$. Due to the small floater pitching angle

assumption, $\xi_5 \ll 1$, the term $V_{0,l} \cos(\xi_5)$ is approximated as $V_{0,l}$.

The spatially uniform turbulent inflow is taken at the hub height of $119 \, \mathrm{m}$ from a Mann turbulence box (Mann, 1994), with a mean inflow velocity at the hub of $V_0 = 8 \, \mathrm{m \, s^{-1}}$ and a mean hub turbulence intensity (TI) of $5.77 \, \%$. The Mann turbulence grid box has a constant spatial step $d\hat{y}$ in the inflow direction, and the time step increment is given by $dt = d\hat{y}/V_0$ (Mann, 1994).

To calculate the impact of the inflow velocity fluctuation on the aerodynamic load $F_{l,aero}$, the velocity triangle components

of the airfoil at the reference radial position ($r = d$) must be investigated. The resulting velocity triangle description is presented in Appendix A.

## 3 Model linearization with forcing

Further, we linearize the model to enable fast response calculations. The linearization methodology of aerodynamic variables, that are velocity dependent, considers a steady term (noted $st$) and a linear variation, as demonstrated here for variable $Y_l$, i.e.

$Y_{l,lin} = Y_{l,st} + \Delta Y_l$. Using a first-order Taylor expansion, the linearized variable $Y_{l,lin}$ is

$$
Y_{l,lin} = Y_{l,st} + \left. \frac{\partial Y_l}{\partial \dot{a}_l} \right|_{st} \dot{a}_l + \left. \frac{\partial Y_l}{\partial f_{s,l}} \right|_{st} f_{s,l} + \left. \frac{\partial Y_l}{\partial \dot{\xi}_5} \right|_{st} \dot{\xi}_5 + \left. \frac{\partial Y_l}{\partial \Delta V_{0,l}} \right|_{st} \Delta V_{0,l},
\tag{8}
$$

where we linearize with respect to the time derivative of the structural DOFs vector, $\dot{x}$, the dynamic stall separation function DOFs, $f_{s,l}$, and the inflow velocity fluctuation, $\Delta V_{0,l}$. In addition, the partial derivatives concerning the inflow angle $\phi_l$ play a significant role in the linearization of the system equations and that is detailed in Appendix A.



### 3.1 Dynamic stall

The Øye dynamic stall model (Øye, 1991) that is implemented in this paper only serves as a fast response methods demonstration and is not meant to be compared to experimental data. In Øye's model, dynamic stall is captured in the lift coefficient $C_L$ via the flow separation function $f_s$, which represents the location $x$ of the trailing edge separation point measured from the leading edge, normalized by the chord length, meaning that $f_s = x/c$ (Øye, 1991). Within this framework, $C_{L,\mathrm{inv}}(\alpha)$ refers to the lift coefficient under inviscid or fully attached flow ($f_s = 1$), while $C_{L,\mathrm{stall}}(\alpha)$ pertains to a fully separated flow ($f_s = 0$) with stall occurring at the leading edge. The Øye stall model equations are presented in Appendix B, where we also justify the choice of this stall model for the present work, supported by our previous studies (Pamfil et al., 2025).

### 3.2 Aerodynamic loads linearization

To obtain the LM EOM, the aerodynamic force $F_{l,aero}$ from Eq. (A1) can be linearized through

$$
\frac{\partial F_{l,lin}}{\partial \cdot} = \frac{\partial \left( L_{l,lin} \cos \phi_{l,lin} \right)}{\partial \cdot} = \frac{1}{2} \rho c \left( \left. \frac{\partial C_{L,l}}{\partial \cdot} \right|_{st} \cos \phi_{st} V_{rel,st}^2 \right.
$$
$$
\left. + C_{L,st} \left. \frac{\partial \cos \phi_{l,lin}}{\partial \cdot} \right|_{st} V_{rel,st}^2 + C_{L,st} \cos \phi_{st} \left. \frac{\partial \left( V_{rel,l}^2 \right)}{\partial \cdot} \right|_{st} \right), \tag{9}
$$

where the system linearized equations have to be considered. The linearization methodology serves to obtain the linearized EOM,

$$
\mathbf{M}_S \ddot{\boldsymbol{x}} + (\mathbf{C}_S + \mathbf{C}_A) \dot{\boldsymbol{x}} + \mathbf{K}_S \boldsymbol{x} = \boldsymbol{F}_L, \tag{10}
$$

through the formulation of a linearized forcing vector $\boldsymbol{F}_L$ and an aerodynamic damping matrix contribution that was formerly developed in Eq. (27) from our stability analysis paper (Pamfil et al., 2025). The aerodynamic damping matrix $\mathbf{C}_A$ in the LM EOM is computed at steady-state ($st$) conditions for an operational point. That operational point is characterized by a specific rotational speed, $\Omega$, and a constant inflow velocity, $V_0$, without an inflow velocity variation, $\Delta V_{0,l}$, taken into account. The aerodynamic forcing terms from Eqs. (1) and (2) are linearized with the notation $GF_{a_l,lin} = F_{l,lin} L_b \phi_{1f}(d)$ and $M_{aero,lin} = \sum_{l=1}^{N_b} F_{l,lin} L_b \left( H + d \cos \Psi_l \right)$. The linearized loads $GF_{a_l,lin}$ and $M_{aero,lin}$ can also be derived partially with respect to the separation function $f_s$ to obtain the Jacobian matrix $[\partial \boldsymbol{F}_i / \partial \boldsymbol{f}_{s,j}]$ from the earlier model used for stability analysis purposes, refer to Eq. (32) (Pamfil et al., 2025). In Eq. (32) (Pamfil et al., 2025), which involves the partial derivative $\frac{\partial C_{L,i}}{\partial f_{s,j}}$, the dynamic stall degrees of freedom (DOFs) vector is defined as $\boldsymbol{F}_s = [f_{s,1}, f_{s,2}, f_{s,3}]^T$, and the aerodynamic forcing vector is given by $\boldsymbol{F} = [M_{\mathrm{aero,lin}}, GF_{a_1,\mathrm{lin}}, GF_{a_2,\mathrm{lin}}, GF_{a_3,\mathrm{lin}}]^T$.

### 3.3 State-space representation with forcing input

To assemble a first-order state-space model ODE, we rely first on the EOM of the TDM from Eq. (3), or the LM's EOM from Eq. (10). We combine the EOM, which is a second-order ODE, with either the original dynamic stall first-order ODE or its





fully linearized variant (Eq. (B2)). The resulting state-space model is presented as

$$\dot{\boldsymbol{q}} = \mathbf{A}\,\boldsymbol{q} + \boldsymbol{F}_B. \tag{11}$$

In this expression, the state vector $\boldsymbol{q} = \left[\boldsymbol{x}_{4\times1}^T, \dot{\boldsymbol{x}}_{4\times1}^T, \boldsymbol{f}_s^T\right]^T$ is of length $N_s = 11$ and contains the structural DOFs vector $\boldsymbol{x}$,

its time derivative $\dot{\boldsymbol{x}}$ and the dynamic stall variable $f_{s,l}$ for each blade within $\boldsymbol{f}_s = [f_{s,1}, f_{s,2}, f_{s,3}]^T$. The system matrix $\mathbf{A}$ is respectively developed as follows for the TDM,

$$\mathbf{A}_T = \begin{bmatrix} [\mathbf{0}_{4\times4}] & [\mathbf{I}_{4\times4}] & [\mathbf{0}_{4\times3}] \\ \left[-\mathbf{M}_S^{-1}\mathbf{K}_S\right] & \left[-\mathbf{M}^{-1}\mathbf{C}_S\right] & \left[\mathbf{M}_S^{-1}\left[\partial\boldsymbol{F}_i/\partial\boldsymbol{f}_{s,j}\right]\right] \\ [\mathbf{0}_{3\times4}] & [\mathbf{0}_{3\times4}] & \left[\partial\dot{\boldsymbol{f}}_{s,i}/\partial\boldsymbol{f}_{s,j}\right] \end{bmatrix} \tag{12}$$

and for the LM,

$$\mathbf{A}_L = \begin{bmatrix} [\mathbf{0}_{4\times4}] & [\mathbf{I}_{4\times4}] & [\mathbf{0}_{4\times3}] \\ \left[-\mathbf{M}_S^{-1}\mathbf{K}_S\right] & \left[-\mathbf{M}_S^{-1}\left(\mathbf{C}_S + \mathbf{C}_A\right)\right] & \left[\mathbf{M}_S^{-1}\left[\partial\boldsymbol{F}_i/\partial\boldsymbol{f}_{s,j}\right]\right] \\ [\mathbf{0}_{3\times4}] & \left[\partial\dot{\boldsymbol{f}}_{s,i}/\partial\dot{\boldsymbol{x}}_j\right] & \left[\partial\dot{\boldsymbol{f}}_{s,i}/\partial\boldsymbol{f}_{s,j}\right] \end{bmatrix}_{st}, \tag{13}$$

whose matrix components are evaluated at the steady-state ($st$) for the operational conditions. The linearization of the ODE for $\dot{f}_{s,l}$ in Eq. (B2) is represented by the two Jacobian matrices, $\left[\partial\dot{\boldsymbol{f}}_{s,i}/\partial\dot{\boldsymbol{x}}_j\right]_{3\times4}$ and $\left[\partial\dot{\boldsymbol{f}}_{s,i}/\partial\boldsymbol{f}_{s,j}\right]_{3\times3}$. For these two Jacobian matrices, the element of row index $i$ is partially derived with respect to the variable of column $j$ index. To verify that the LM exhibits a physically consistent behavior, we previously performed decay test simulations with initial perturbations (Pamfil et al., 2025) to compare results against the TDM. The results in Fig. 5 (Pamfil et al., 2025) were expressed as deviations from

the steady-state values and the time-domain plots confirmed the consistency between the results produced by the TDM and the LM. To further elaborate the state-space model, the forcing input vector $\boldsymbol{F}_B$ for the TDM,

$$\boldsymbol{F}_{B,T} = \begin{bmatrix} [\mathbf{0}_{4\times1}] \\ \mathbf{M}_S^{-1} \begin{bmatrix} \sum_{l=1}^3 \boldsymbol{F}_{T,\xi_5}\left(C_{L,stall,l}\right) + M_F \\ \boldsymbol{F}_{T,a_1}\left(C_{L,stall,1}\right) \\ \boldsymbol{F}_{T,a_2}\left(C_{L,stall,2}\right) \\ \boldsymbol{F}_{T,a_3}\left(C_{L,stall,3}\right) \end{bmatrix} \\ \begin{bmatrix} f_{s,static,1}/\tau \\ f_{s,static,2}/\tau \\ f_{s,static,3}/\tau \end{bmatrix} \end{bmatrix}, \tag{14}$$





does not consider a full linearization of the dynamic stall ODE unlike the LM as follows

$$
\boldsymbol{F}_{B,L} = \begin{bmatrix} [\boldsymbol{0}_{4\times1}] \\ \mathbf{M}_S^{-1} \begin{bmatrix} \underbrace{M_F + \sum_{l=1}^{3} \left( \boldsymbol{F}_{L,\xi_5}\left(\Delta V_{0,l}\right) + \boldsymbol{F}_{L,\xi_5}\left(\Delta V_{0,l}^2\right) \right)}_{M_{aero,lin}\left(\Delta V_{0,l}, \Delta V_{0,l}^2\right)} \\ GF_{a_1,lin}\left(\Delta V_{0,1}, \Delta V_{0,1}^2\right) = \boldsymbol{F}_{L,a_1}\left(\Delta V_{0,1}\right) + \boldsymbol{F}_{L,a_1}\left(\Delta V_{0,1}^2\right) \\ GF_{a_2,lin}\left(\Delta V_{0,2}, \Delta V_{0,2}^2\right) = \boldsymbol{F}_{L,a_2}\left(\Delta V_{0,2}\right) + \boldsymbol{F}_{L,a_2}\left(\Delta V_{0,2}^2\right) \\ GF_{a_3,lin}\left(\Delta V_{0,3}, \Delta V_{0,3}^2\right) = \boldsymbol{F}_{L,a_3}\left(\Delta V_{0,3}\right) + \boldsymbol{F}_{L,a_3}\left(\Delta V_{0,3}^2\right) \end{bmatrix} \\ \begin{bmatrix} \frac{1}{\tau}\left( \left.\frac{\partial f_{s,static,1}}{\partial \alpha_1}\right|_{st} \left.\frac{\partial \phi_1}{\partial \Delta V_{0,1}}\right|_{st} \Delta V_{0,1} \right) \\ \frac{1}{\tau}\left( \left.\frac{\partial f_{s,static,2}}{\partial \alpha_2}\right|_{st} \left.\frac{\partial \phi_2}{\partial \Delta V_{0,2}}\right|_{st} \Delta V_{0,2} \right) \\ \frac{1}{\tau}\left( \left.\frac{\partial f_{s,static,3}}{\partial \alpha_3}\right|_{st} \left.\frac{\partial \phi_3}{\partial \Delta V_{0,3}}\right|_{st} \Delta V_{0,3} \right) \end{bmatrix} \end{bmatrix}.
\tag{15}
$$

The forcing vector from the EOM has components for both the TDM (index $T$) and LM (index $L$) that pertain to structural DOFs, and to the aerodynamic DOFs by taking into account the Øye dynamic stall model. The LM's forcing input $\boldsymbol{F}_{B,L}$ explicitly accounts for variations in aerodynamic forcing parameters and dynamic stall with respect to the per-blade inflow velocity change, $\Delta V_{0,l}$. To this end, the LM solution for the state vector $\boldsymbol{q}$ is computed using a fourth-order Runge-Kutta (RK4) method with a fixed time step interval $dt$.

## 4    Fast response methods and Hill's decomposition

The response $\boldsymbol{q}$ from the state-space model shown in Eq. (11) can be solved in the time domain directly. Yet it can be beneficial to solve it using the fast response calculation methodology in the frequency domain or the Laplace $s$-domain since the state-space Eq. (11) describes a linear system. An advantage of a Laplace transform-based approach is that it captures accurately the transient response due to the consideration of initial conditions, just like for the TDM and LM. Conversely, the fast response

methods using the Fourier transform neglect transient response effects from initial conditions.

Using the LM, the numerical procedures that are presented in this paper can be utilized for different load cases and for multiple floating wind turbine configurations such as with variable rotors (symmetric or asymmetric, different number of blades, isotropic or anisotropic blades), or variable floater types (spar-buoy, semi-submersible, damping pool floating foundation). The distinctions in simulations parameters can be as well in terms of the dynamic stall model that is considered or with the inclusion

of a controller in the state-space model. All of these different simulation conditions would affect the LM system matrix $\mathbf{A}_L$ and the forcing vector $\boldsymbol{F}_L$.

To derive a Fourier transform-based fast response calculation procedure, the forcing term vector time series, $\boldsymbol{F}_B(t)$, of a duration equal to the simulation time $T_{sim}$, is converted to the frequency domain. Similarly to the variable $X(t)$, both the forcing term vector $\boldsymbol{F}_B(t)$ and the state variable $\boldsymbol{q}(t)$ are expressed in the frequency domain as $X(t) = \sum_{s=-N}^{N} X(\omega_s)e^{\mathrm{i}\omega_s t}$,





where the frequency samples are defined as $\omega_s = \frac{2\pi s}{T_{\text{sim}}}$ and $\mathrm{i} = \sqrt{-1}$ is the imaginary unit. This conversion is carried out in order to compute the response $q$ in the frequency domain.

We then utilize the FFT algorithm to obtain the frequency-domain forcing $F_B(\omega)$. The FFT-based methods, however, assume that the time signal is periodic within the simulation time frame. When this assumption does not apply, windowing functions are employed to remove the time signal edge effects. With this objective, we impose a window function $W(t)$ on the time series of the LM input forcing vector $F_{B,L}(t)$, $\boldsymbol{F}_{B,L}(\omega) = \mathrm{FFT}\left(\boldsymbol{F}_{B,L}(t) \cdot W(t)\right)$. The multiplication of the forcing input by the window function as $F_{B,L}(t) \cdot W(t)$ serves to mitigate the effects of spectral leakage or aliases. The window function $W(t) = R_1(t) \cdot R_2(t)$, as illustrated in Fig. 2, is symmetric in time with the two ramp functions given by $R_1(t) = \tanh^2\left(\frac{t}{f_{ramp}T_{\xi_5}}\right)$ and $R_2(t) = \tanh^2\left(\frac{T_{sim}-t}{f_{ramp}T_{\xi_5}}\right)$. They are time scaled by a ramping factor chosen as $f_{ramp} = 2$ and by the largest natural period of the system that corresponds to the floater pitch natural period, $T_{\xi_5} = 2\pi/\omega_{\xi_5}$.

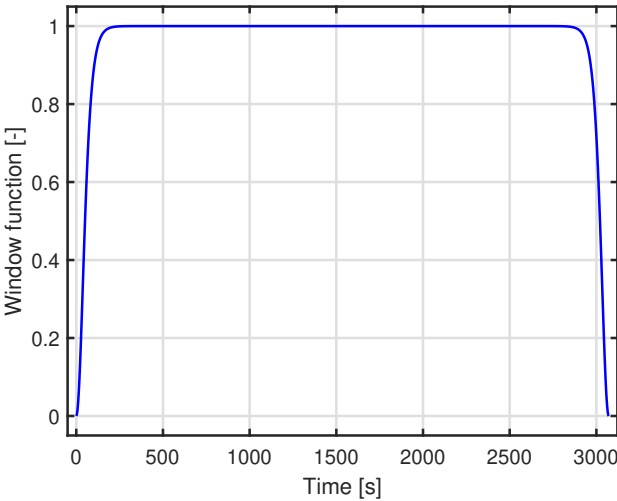

**Figure 2.** Window function $W(t) = R_1(t) \cdot R_2(t)$ for a simulation duration of $T_{sim} = 3071.2$ s.

One point worth mentioning is that through the application of the window function, the time span is reduced where the fast response results can be compared to the benchmark results. As can be seen in Fig. 2, the time lapse valid for comparison between results is located where the window function is equal to $1$. The simulation time lost due to windowing (i.e., the portion valid for analysis) is smaller than the time lost due to the initial transient period.

Furthermore, the governing state-space Eq. (11) can be expressed in the frequency domain by converting the time-sampled terms into their frequency-domain representations, yielding

$$\sum_{s=-N}^{N} \mathrm{i}\omega_s \boldsymbol{q}(\omega_s) e^{\mathrm{i}\omega_s t} = \sum_{s=-N}^{N} \left([\mathbf{A}\,\boldsymbol{q}]_{\omega_s} + \boldsymbol{F}_B(\omega_s)\right) e^{\mathrm{i}\omega_s t}. \tag{16}$$

The cut-off frequency for the truncation in Eq. (16) must be sufficiently large to cover the wind spectrum. If matrix $\mathbf{A}$ is constant, then after simplification of Eq. (16) we get $(\mathrm{i}\omega\mathbf{I} - \mathbf{A})\,\boldsymbol{q}(\omega) = \boldsymbol{F}_B(\omega)$. Thus, the response vector $\boldsymbol{q}(\omega)$ can be computed



according to,

$$q(\omega) = \underbrace{(\mathrm{i}\omega\mathbf{I} - \mathbf{A})^{-1}}_{\mathbf{H}(\omega)} \boldsymbol{F}_B(\omega), \tag{17}$$

which defines the transfer function matrix $\mathbf{H}(\omega)$. The core principle of fast response calculations is to convert the response solution from the frequency domain to the time domain using the inverse Fast Fourier Transform (iFFT) algorithm, $q(t) = \Re\{\mathrm{iFFT}(q(\omega))\}$, and then extracting only the real part of the result. The frequency-domain response signal $q(\omega)$ must be padded, which involves adding zeros to extend the data to the full length of the original time series before applying the inverse FFT.

To include the contribution of higher harmonics in the response $q(t)$, we take inspiration in Hill's decomposition method for stability analysis of periodic systems and apply a double-sided Fourier decomposition in $\mathbf{A}_L(t)$ and $q(t)$. Hill's decomposition is related to previous studies that we performed for response (Pamfil et al., 2024) and stability (Pamfil et al., 2025) analyzes.

Hill's decomposition requires first the double-sided Fourier decomposition of time-periodic quantities such as $\mathbf{A}_L(t)$ and $q(t)$, which in general can be expressed for any periodic variable $\mathbf{X}(t)$ as $\mathbf{X}(t) = \sum_{n=-\infty}^{\infty} \mathbf{X}_n e^{\mathrm{i}n\Omega t}$. In that case, each matrix $\mathbf{A}_{L,n}$ is constant and both variables $q(t)$ and $\mathbf{A}_L(t)$ are of dimension $N_s$. The frequency components $\mathbf{X}_n$ of a time-dependent variable $\mathbf{X}(t)$, such as $\mathbf{A}_{L,n}$ for $\mathbf{A}_L(t)$, can be obtained using the numerical trapezoidal integration method, i.e. $\mathbf{X}_n = \frac{1}{T}\int_0^T \mathbf{X}(t)e^{-\mathrm{i}n\Omega t}dt$, where $T = 2\pi/\Omega$ is the period of the system for a given constant rotational speed $\Omega$. Alternatively, since there are sufficient time samples over one period $T$, the FFT algorithm can be used efficiently. The Fourier decomposition is defined as double-sided, ensuring that the system matrix of the LM, $\mathbf{A}_L(t)$, the state vector $q$, and its time derivative $\dot{q}$ remain purely real. This is achieved by the cancellation of the imaginary components arising from the positive $(+n\Omega)$ and negative $(-n\Omega)$ harmonics, with a truncation upper limit of $N = 4$ in this case study, ensuring an accurate Fourier expansion of $\mathbf{A}_L(t)$. As explained in our previous study (Pamfil et al., 2025), the Fourier decomposition of the system matrix $\mathbf{A}_L(t)$ from Eq. (13) is valid due to the azimuthal periodicity of the system at a fixed rotational speed $\Omega$. This Fourier decomposition of the state-space variables for the free vibration case was introduced by Hill (1886) and it is commonly used for a stability analysis. Hill's theory allows to carry out a stability analysis of a floating wind turbine while taking into account the rotor's periodicity. For the unforced problem ($\boldsymbol{F}_B = \mathbf{0}$ in Eq. (11)), the periodic eigenmodes $\psi_k$ with principle eigenvector components $\hat{\boldsymbol{v}}_{k,n}$, $\psi_k = \sum_{n=-N}^{N} \hat{\boldsymbol{v}}_{k,n} e^{\mathrm{i}n\Omega t}$, are found by substitution of the solution $\psi_{k,sol}(t) = \psi_k e^{\lambda_k t}$, which considers a principal eigenvalue $\lambda_k$ per mode. This eigenvalue problem leads to the hyper-matrix LTI formulation for the unforced case given by $\dot{\hat{q}} = \hat{\mathbf{A}}\hat{q}$:

$$\begin{bmatrix} \vdots \\ \dot{q}_{n=-1} \\ \dot{q}_{n=0} \\ \dot{q}_{n=1} \\ \vdots \end{bmatrix} = \begin{bmatrix} \ddots & \vdots & \vdots & \vdots & \iddots \\ \cdots & \mathbf{A}_{L,0}+\mathrm{i}\Omega\mathbf{I} & \mathbf{A}_{L,-1} & \mathbf{A}_{L,-2} & \cdots \\ \cdots & \mathbf{A}_{L,1} & \mathbf{A}_{L,0} & \mathbf{A}_{L,-1} & \cdots \\ \cdots & \mathbf{A}_{L,2} & \mathbf{A}_{L,1} & \mathbf{A}_{L,0}-\mathrm{i}\Omega\mathbf{I} & \cdots \\ \iddots & \vdots & \vdots & \vdots & \ddots \end{bmatrix} \begin{bmatrix} \vdots \\ q_{n=-1} \\ q_{n=0} \\ q_{n=1} \\ \vdots \end{bmatrix}, \tag{18}$$

which we have clarified in our work on floating wind turbine stability analysis (Pamfil et al., 2025). That being said, we have proven (Pamfil et al., 2025) that the formulation of the hyper-matrix $\hat{\mathbf{A}}$ can be found by replacing the Fourier decomposed terms





$q(t)$ and $\mathbf{A}_L(t)$ into the state-space Eq. (11) and rearranging them, for derivations refer to Eqs. (42), (43), and (44) within our previous paper (Pamfil et al., 2025). Hence, by varying the index $n$ from integer $-N$ to $N$ in Eq. (44) (Pamfil et al., 2025), the row equations of the hyper-matrix expression can be constructed. It should be noted that the harmonic matrices $\mathbf{A}_{L,n}$ required for assembling $\hat{\mathbf{A}}$ extend from $\mathbf{A}_{L,n=-2N}$ to $\mathbf{A}_{L,n=2N}$. Moreover, under the assumption that the lower harmonics of matrix $\mathbf{A}_L(t)$ are greater than the higher harmonics,

$$\|\mathbf{A}_{L,0}\|_F > \|\mathbf{A}_{L,\pm 1}\|_F > \|\mathbf{A}_{L,\pm 2}\|_F...,\tag{19}$$

the constant hyper-matrix $\hat{\mathbf{A}}$ can be truncated and its eigenvalues would still be accurate.

Although Eq. (18) is derived for the unforced system, it suggests that the original LM may be recast into an LTI system with the hyper-matrix as a constant system matrix. One would need, however, to add the forcing term $\boldsymbol{F}_{B,L}(t)$ which will generally have frequency content beyond the harmonics of the rotational frequency $\Omega$. Even though the central row in Eq. (18) appears to be the natural position for the forcing term $\boldsymbol{F}_{B,L}(t)$, this requires a formal assumption. In a previous study that we conducted (Pamfil et al., 2024), for a forced response excitation with a more simplified floating wind turbine model, this approach was tested with a corresponding forcing hyper-vector $\hat{\boldsymbol{F}}_{B,L}(t) = [...,\boldsymbol{0}^T, \boldsymbol{F}_{B,L}^T(t), \boldsymbol{0}^T,...]^T$ (Pamfil et al., 2024). That produced identical results to the TDM that was linearized which can be confirmed by inspecting time series in Figs. 6, 7, 16, 17, 18 and 19 (Pamfil et al., 2024). Hence, in Section 5, we utilize Eq. (19) to formalize the perturbation methods for fast response computations, and de-couple Eq. (18) into smaller sub-systems.

To develop novel fast response methods, we take inspiration in Hill's decomposition and assemble a single-sided Fourier series by grouping together the positive and negative harmonic components of the same harmonic order in absolute value. The variables $e^{in\Omega t}$, $\boldsymbol{q}_n$, and $\mathbf{A}_{L,n}$ are combined as single terms within the single-sided Fourier decomposition through their complex conjugates (noted $\cdot^*$). When considering the complex conjugate of a harmonic term $X_n$, being $X_n^*$, it is known that $X_{-n} = X_n^*$, which conveys that $\Re\{X_{-n}\} = \Re\{X_n\}$ and $\Im\{X_{-n}\} = -\Im\{X_n\}$. We apply this notion to obtain single-sided Fourier decompositions of $\boldsymbol{q}(t)$ and $\mathbf{A}_L(t)$. Each harmonic component now depends on the real ($\Re\{\cdot\}$) and imaginary ($\Im\{\cdot\}$) parts for variables $e^{in\Omega t}$, $\boldsymbol{q}(t)$ and $\mathbf{A}_L(t)$. This is outlined for the term $X(t)$ which represents either variable $\boldsymbol{q}(t)$ or $\mathbf{A}_L(t)$:

$$\begin{aligned}X(t) &= \tilde{X}_0 + \tilde{X}(t)\\ &= \tilde{X}_0 + \sum_{n=1}^{N}\tilde{X}_n(t)\\ &= \tilde{X}_0 + \sum_{n=1}^{N}\left(X_n e^{in\Omega t} + X_{-n}e^{-in\Omega t}\right)\\ &= \tilde{X}_0 + \sum_{n=1}^{N}2\left(\Re\{X_n\}\Re\{e^{in\Omega t}\} - \Im\{X_n\}\Im\{e^{in\Omega t}\}\right)\\ &= \tilde{X}_0 + \sum_{n=1}^{N}\left(X_{n,c}\cos(n\Omega t) + X_{n,s}\sin(n\Omega t)\right).\end{aligned}\tag{20}$$

In Eq. (20), the average term $X_0$ is written as $\tilde{X}_0$ instead to ensure consistency in notation. Since the term $X(t)$ is purely real, Eq. (20) demonstrates that the double-sided complex Fourier series (exponential form) can also be expressed as the single-sided





real Fourier series (sine-cosine form). The exponential form of the Fourier series translates to a sine-cosine form, where, for
each index $n$, the real cosine and sine coefficients $X_{n,c}$ and $X_{n,s}$ are associated with the complex coefficient $X_n$.

## 5 Perturbation methods

We develop fast response methods by relying on the Hill expansion. We recast the Hill expansion instead as a Taylor expansion
with a small formal ordering parameter $\delta^n$ which multiplies the corresponding harmonic terms of order $n$:

$$
\begin{aligned}
\boldsymbol{q}(t) &= \sum_{n=-N}^{N} \delta^{|n|} \boldsymbol{q}_n(t) e^{in\Omega t} = \tilde{\boldsymbol{q}}_0(t) + \sum_{n=1}^{N} \delta^n \tilde{\boldsymbol{q}}_n(t) \\
\mathbf{A}_L(t) &= \sum_{n=-N}^{N} \delta^{|n|} \mathbf{A}_{L,n} e^{in\Omega t} = \tilde{\mathbf{A}}_{L,0} + \sum_{n=1}^{N} \delta^n \tilde{\mathbf{A}}_{L,n}(t).
\end{aligned}
\tag{21}
$$

As done in Eq. (20), the negative and positive harmonic terms from the double-sided Fourier expansion, $X_{-n}e^{-in\Omega t}$ and
$X_n e^{in\Omega t}$, are combined together as terms of the same harmonic order within a single-sided Fourier series. According to the
perturbation method (Bender and Orszag, 1999), the harmonic ordering is explicitly carried out through a perturbative decom-
position, $X(t) = \tilde{X}_0 + \sum_{n=1}^{N} \delta^n \tilde{X}_n(t)$, where the harmonic terms originate from the single-sided Fourier series. The higher
harmonics response contributions $\tilde{\boldsymbol{q}}_n(t)$ are solved up to a desired order $n$. In the following Sections 5.1 and 5.2, we will
present the double and single perturbation methods to achieve this.

### 5.1 Double perturbation method

Our first perturbation method is obtained through insertion of the perturbation expansions from Eq. (21) into the state-space
model from Eq. (11). After applying this perturbative decomposition to the state-space ODE terms we get:

$$
\sum_{n=0}^{N} \dot{\tilde{\boldsymbol{q}}}_n \delta^n = \sum_{p=0}^{N} \sum_{j=0}^{N} \tilde{\mathbf{A}}_{L,j} \, \tilde{\boldsymbol{q}}_p \delta^{(p+j)} + \boldsymbol{F}_{B,L}.
\tag{22}
$$

The Hill decomposition of the matrix $\mathbf{A}_L$ into its harmonics $\mathbf{A}_{L,j}$ can be performed without needing to extract the harmonics
from the hyper-matrix $\hat{\mathbf{A}}$.

Concerning the LM forcing input $\boldsymbol{F}_{B,L}$, it is only associated to the unit perturbation of $\delta^0 = 1$ which is linked to the zeroth
harmonic order.

Continuing from Eq. (22), we can isolate each $n^{th}$ set of equations of the same order of magnitude $\delta^n$. We can identify the
zeroth harmonic equation through the zeroth perturbation order $\delta^0$ as shown in Eq. (23). The zeroth harmonic response $\tilde{\boldsymbol{q}}_0$ can
be calculated through the transfer function $\mathbf{H}(\omega)$, see Eq. (17).

Bir (2008) has shown that considering only the averaged system matrix $\tilde{\mathbf{A}}_{L,0}$ over a period means neglecting periodic terms
that can contribute to the system dynamics. As we supported with a more simplified floating wind turbine model (Pamfil et al.,
2024), for some load cases, the zeroth order response, $\tilde{\boldsymbol{q}}_0$, is insufficient to account for the total response $\boldsymbol{q}$ when the latter
is highly periodic. We have demonstrated (Pamfil et al., 2024) that the zeroth harmonic state response $\tilde{\boldsymbol{q}}_0$ is also equivalent



to solving in the frequency domain the zeroth order structural DOF vector $\tilde{x}_0(\omega)$ from the LM EOM (Eq. (10)) with zeroth order mass, damping and stiffness matrices. Using the LM EOM is the conventional way of solving the floating wind turbine response in the frequency domain. Consequently, based on Eq. (22), we build a system of equations to solve consecutively higher-order contributions in the following manner:

$$
\begin{aligned}
\delta^0 : \quad & \dot{\tilde{q}}_0(t) = \tilde{\mathbf{A}}_{L,0}\tilde{q}_0 + \boldsymbol{F}_{B,L} \\
\delta^1 : \quad & \dot{\tilde{q}}_1(t) = \tilde{\mathbf{A}}_{L,0}\tilde{q}_1 + \tilde{\mathbf{A}}_{L,1}\tilde{q}_0 \\
& \vdots \\
\delta^n : \quad & \dot{\tilde{q}}_n(t) = \sum_{j=0}^{n} \tilde{\mathbf{A}}_{L,j}\,\tilde{q}_{n-j} = \tilde{\mathbf{A}}_{L,0}\tilde{q}_n + \sum_{j=1}^{n} \tilde{\mathbf{A}}_{L,j}\,\tilde{q}_{n-j}, \quad n > 0.
\end{aligned}
\tag{23}
$$

After solving the zeroth harmonic response, $\tilde{q}_0$, the sequential solving strategy from Eq. (23) is implemented for higher-order harmonic ($n > 0$) responses $\tilde{q}_n$. It can be expressed in a lower triangular hyper-matrix form,

$$
\begin{bmatrix} \dot{\tilde{q}}_0 \\ \dot{\tilde{q}}_1 \\ \dot{\tilde{q}}_2 \\ \vdots \end{bmatrix}
=
\begin{bmatrix}
\tilde{\mathbf{A}}_{L,0} & \mathbf{0} & \mathbf{0} & \dots \\
\tilde{\mathbf{A}}_{L,1} & \tilde{\mathbf{A}}_{L,0} & \mathbf{0} & \dots \\
\tilde{\mathbf{A}}_{L,2} & \tilde{\mathbf{A}}_{L,1} & \tilde{\mathbf{A}}_{L,0} & \dots \\
\vdots & \vdots & \vdots & \ddots
\end{bmatrix}
\begin{bmatrix} \tilde{q}_0 \\ \tilde{q}_1 \\ \tilde{q}_2 \\ \vdots \end{bmatrix}
+
\begin{bmatrix} \boldsymbol{F}_{B,L} \\ \mathbf{0} \\ \mathbf{0} \\ \vdots \end{bmatrix}.
\tag{24}
$$

Eq. (24) would be solved through a forward substitution similarly to the iterative method of Gauss–Seidel with successive displacement. This iterative solving protocol is identical to the double perturbation method presented in Eq. (23). While the original linear problem expressed through Eqs. (11) and (21) corresponds in Eq. (23) to the sum of the equations in one operation, the perturbation approach breaks this in smaller sub-problems which are solved sequentially.

In summary, the higher-order harmonic responses $\tilde{q}_n$ ($n > 0$) are computed successively in the frequency domain (see Eq. (17)) as follows

$$
\tilde{q}_n(\omega) = \underbrace{\left( \mathrm{i}\omega\mathbf{I} - \tilde{\mathbf{A}}_{L,0} \right)^{-1}}_{\mathbf{H}(\omega)} \sum_{j=1}^{n} \tilde{\mathbf{A}}_{L,j}\tilde{q}_{n-j}(\omega),
\tag{25}
$$

where $\mathbf{H}(\omega)$ is the transfer function and $\sum_{j=1}^{n} \tilde{\mathbf{A}}_{L,j}\tilde{q}_{n-j}(\omega)$ is the numerical forcing term. In the end, we calculate the full response by summing all response harmonics according to Eq. (21), and by converting the solution to the time domain via the iFFT algorithm.

## 5.2 Single perturbation method

As an alternative to the perturbative expansion in Eq. (21), the system matrix $\mathbf{A}_L(t)$ can be expressed as a zeroth- and first-order perturbation of $\varepsilon$, encompassing all higher harmonic contributions, such that $\mathbf{A}_L(t) = \tilde{\mathbf{A}}_{L,0} + \varepsilon \tilde{\mathbf{A}}_L(t)$. The small perturbation of $n^{th}$ order, $\varepsilon^n$, is applied to the response $\boldsymbol{q}(t)$ harmonics and to its time derivative $\dot{\boldsymbol{q}}(t)$ harmonics. That is equivalent to





the double perturbation approach applied to $\boldsymbol{q}(t)$ and $\dot{\boldsymbol{q}}(t)$ which results in $\boldsymbol{q}(t) = \tilde{\boldsymbol{q}}_0 + \sum_{n=1}^{N} \varepsilon^n \tilde{\boldsymbol{q}}_n$. The insertion of these perturbation expressions for $\mathbf{A}_L(t)$ and $\boldsymbol{q}(t)$ into Eq. (11) yields

$$
\quad \dot{\tilde{\boldsymbol{q}}}_0 + \sum_{n=1}^{N} \varepsilon^n \dot{\tilde{\boldsymbol{q}}}_n = \boldsymbol{F}_{B,L} + \left( \tilde{\mathbf{A}}_{L,0} + \varepsilon \tilde{\mathbf{A}}_L \right) \left( \tilde{\boldsymbol{q}}_0 + \sum_{n=1}^{N-1} \varepsilon^n \tilde{\boldsymbol{q}}_n \right). \tag{26}
$$

The cumulative contribution of higher-order harmonics terms is expanded to identify terms for each power of $\varepsilon$. That gives the following sequence of equations that can each be solved through the transfer function $\mathbf{H}(\omega)$ as in Eq. (25):

$$
\begin{aligned}
\varepsilon^0 : \quad & \dot{\tilde{\boldsymbol{q}}}_0(t) = \tilde{\mathbf{A}}_{L,0} \tilde{\boldsymbol{q}}_0 + \boldsymbol{F}_{B,L} \\
\varepsilon^1 : \quad & \dot{\tilde{\boldsymbol{q}}}_1(t) = \tilde{\mathbf{A}}_{L,0} \tilde{\boldsymbol{q}}_1 + \tilde{\mathbf{A}}_L \tilde{\boldsymbol{q}}_0 \\
& \vdots \\
\varepsilon^n : \quad & \dot{\tilde{\boldsymbol{q}}}_n(t) = \tilde{\mathbf{A}}_{L,0} \tilde{\boldsymbol{q}}_n + \tilde{\mathbf{A}}_L \tilde{\boldsymbol{q}}_{n-1}, \quad n > 0.
\end{aligned}
\tag{27}
$$

In contrast to the double perturbation method, the decomposition of $\mathbf{A}_L(t)$ can be achieved without a full Hill expansion, because $\tilde{\mathbf{A}}_{L,0}$ can be calculated by averaging $\mathbf{A}_L(t)$ over one period $T = 2\pi/\Omega$ and $\tilde{\mathbf{A}}_L(t) = \mathbf{A}_L(t) - \tilde{\mathbf{A}}_{L,0}$.

Just like for the double perturbation, additional insight can be gained by observing that the single perturbation method in Eq. (27) can be expressed as well by a lower triangular hyper-matrix formulation,

$$
\begin{bmatrix} \dot{\tilde{\boldsymbol{q}}}_0 \\ \dot{\tilde{\boldsymbol{q}}}_1 \\ \dot{\tilde{\boldsymbol{q}}}_2 \\ \vdots \end{bmatrix} = \begin{bmatrix} \tilde{\mathbf{A}}_{L,0} & \mathbf{0} & \mathbf{0} & \ldots \\ \tilde{\mathbf{A}}_L & \tilde{\mathbf{A}}_{L,0} & \mathbf{0} & \ldots \\ \mathbf{0} & \tilde{\mathbf{A}}_L & \tilde{\mathbf{A}}_{L,0} & \ldots \\ \vdots & \vdots & \vdots & \ddots \end{bmatrix} \begin{bmatrix} \tilde{\boldsymbol{q}}_0 \\ \tilde{\boldsymbol{q}}_1 \\ \tilde{\boldsymbol{q}}_2 \\ \vdots \end{bmatrix} + \begin{bmatrix} \boldsymbol{F}_{B,L} \\ \mathbf{0} \\ \mathbf{0} \\ \vdots \end{bmatrix}, \tag{28}
$$

where the responses $\tilde{\boldsymbol{q}}_n$ can be solved similarly to the Gauss-Seidel method of successive displacement (forward substitution).

Finally, the solution of Eq. (27) is computed sequentially in the frequency domain the same way as for the double perturbation method. Once all response harmonics have been computed they are summed together.

## 5.3 Laplace transform

An alternative to calculating the system response in the frequency domain using the FFT algorithm is to compute it in the Laplace $s$-domain using the Laplace transform. To calculate the response in the $s$-domain the system is assumed to be LTI and the state-space ODE from Eq. (11) is analytically transformed into an algebraic equation. Furthermore, the Laplace method proceeds in the same manner as the Fourier transform, to solve the system response in the new domain, and then apply the inverse of the transform to convert it back to the time domain. However, for high-order systems or those with multiple inputs and outputs, performing an analytical symbolic inversion of the Laplace transform can be challenging or impractical, frequently requiring the application of numerical inversion methods. The benefit, in comparison with the Fourier transform method, is that it is capable of considering the initial conditions and transient response, such as for decay tests. The initial condition $\boldsymbol{q}(t=0)$ is taken into account through a time step looping computation procedure where the current time step $t_i$ response is calculated





using the previous time step $t_{i-1}$, and there is a very small constant time step increment $dt$. This approach does simultaneously solve well the non-transient response too.

Using Eq. (27), we apply the single perturbation approach by carrying out the sum of harmonics response results, i.e. $\tilde{\boldsymbol{q}} = \sum_{n=0}^{N} \tilde{\boldsymbol{q}}_n$, only up to the first-order harmonic $\tilde{\boldsymbol{q}}_1$. As a starting point, the zeroth harmonic expression that is found in Eq. (27) can be converted to the $s$-domain. This conversion is carried out through the Laplace transform applied on the left- and right-hand side,

$$\mathcal{L}\left\{\dot{\tilde{\boldsymbol{q}}}_0(t)\right\} = \tilde{\mathbf{A}}_{L,0}\,\mathcal{L}\{\tilde{\boldsymbol{q}}_0(t)\} + \mathcal{L}\{\boldsymbol{F}_{B,L}(t)\} \quad t \in [t_{i-1}, t_i]. \tag{29}$$

The Laplace transform in Eq. (29) is applied locally at each time step $t_i$, over the time interval $[t_{i-1}, t_i]$, which has a duration

equal to the time step interval $dt$. This suggests that the initial condition for that time interval is taken as $\tilde{\boldsymbol{q}}_0(t_{i-1})$, and $\boldsymbol{F}_{B,L}(t_i)$ is assumed to be constant during that time interval. After applying the Laplace transform in Eq. (29), it equates to

$$s\,\tilde{\boldsymbol{q}}_0(s) - \tilde{\boldsymbol{q}}_0(t_{i-1}) = \tilde{\mathbf{A}}_{L,0}\,\tilde{\boldsymbol{q}}_0(s) + \underbrace{\frac{1}{s}\boldsymbol{F}_{B,L}(t_i)}_{\boldsymbol{F}_{B,L}(s)}. \tag{30}$$

The $s$-domain contains a real part $\sigma$ and an imaginary part $\mathrm{i}\omega$, resulting in $s = \sigma + \mathrm{i}\omega$. The Fourier transform is a particular case of the bilateral Laplace transform where the initial conditions are neglected, i.e. $s = \mathrm{i}\omega$ and $\sigma = 0$.

Based on Eq. (30), $\tilde{\boldsymbol{q}}_0(s)$ can be isolated:

$$\mathcal{L}^{-1}\left\{\tilde{\boldsymbol{q}}_0(s)\right\} = \mathcal{L}^{-1}\left\{\left(s\mathbf{I} - \tilde{\mathbf{A}}_{L,0}\right)^{-1}\left(\frac{1}{s}\boldsymbol{F}_{B,L}(t_i) + \tilde{\boldsymbol{q}}_0(t_{i-1})\right)\right\}. \tag{31}$$

The inverse of the Laplace transform is applied to solve the state response $\mathcal{L}^{-1}\{\tilde{\boldsymbol{q}}_0(s)\} = \tilde{\boldsymbol{q}}_0(t_i)$ similarly to the inverse of the FFT for the previous fast response methods. The transfer function $\mathbf{H}(s) = \left(s\mathbf{I} - \tilde{\mathbf{A}}_{L,0}\right)^{-1}$ in Eq. (31) is similar to the transfer function for the frequency domain from Eq. (25). The inverse of the Laplace transform is applied to Eq. (31) so that it can be

solved only once through a symbolic solver, such as MATLAB's symbolic inverse Laplace function `ilaplace`. The terms that are a function of the current time step $t_i$ and previous time step $t_{i-1}$ are treated as constants when solving symbolically Eq. (31). Also, when solving Eq. (31) numerically at each time step, the time variable $t$ is replaced by the time step increment $dt$.

    Afterwards, the same strategy as in Eq. (31) is applied, with a change of variables, to solve the first-order harmonic response

$\tilde{\boldsymbol{q}}_1(t_i)$ at time step $t_i$:

$$\mathcal{L}^{-1}\left\{\dot{\tilde{\boldsymbol{q}}}_1(s)\right\} = \mathcal{L}^{-1}\left\{\left(s\mathbf{I} - \tilde{\mathbf{A}}_{L,0}\right)^{-1}\left(\frac{1}{s}\tilde{\mathbf{A}}_L\boldsymbol{q}_0(t_i) + \tilde{\boldsymbol{q}}_1(t_{i-1})\right)\right\}. \tag{32}$$

We establish the initial condition ($t = 0$) for Eq. (32) that $\tilde{\boldsymbol{q}}_1(t_1) = \boldsymbol{0}$ for the first time step, since $\tilde{\boldsymbol{q}}_0(t_1) = \boldsymbol{q}(t = 0)$. Further, as given by Eq. (21), the response $\tilde{\boldsymbol{q}}_1(t)$ is added to the zeroth harmonic solution $\tilde{\boldsymbol{q}}_0(t)$ to obtain the system response $\boldsymbol{q}(t)$.

    Due to the computationally expensive time iteration procedure, we settle for an accuracy going up only to the first-order

harmonic $\tilde{\boldsymbol{q}}_1(t)$ response. Given the time loop nature of this method, we found out that it was not fast, but had a comparable CPU time as the standard time integration of the LM state-space. Hence, we benchmark the CPU time and accuracy in the following Sections. More details about the computational efficiency are explained in Section 7.





## 6 Results from fast response methods

We present the fast response methods time series and Power Spectral Density (PSD) results for various load cases for the operational point of $V_0 = 8\,\mathrm{m\,s^{-1}}$ and $\Omega = 0.6\,\mathrm{rad\,s^{-1}}$. Since this study does not take into account controller implementation, we select arbitrarily an operational point below the rated velocity, such as $V_0 = 11.4\,\mathrm{m\,s^{-1}}$. This operational condition is used in this entire paper for all simulations load cases. Results are compared between the TDM, the time-dependent LM, the Fourier-based fast response methods and the Laplace-based method. For the Fourier-based and Laplace-based methods, the zeroth harmonic solution serves as the starting point, and its accuracy is increased through the single and double perturbation (Pert.) approaches, with accuracies reaching up to second harmonic order: $O(\varepsilon^2)$ and $O(\delta^2)$ respectively. The TDM results serve as a benchmark to validate if the LM is in accordance with what would be expected, but not as a means to evaluate the accuracy of the fast response and Laplace-based results which should be compared rather with the LM. It has been proven in our previous investigations (Pamfil et al., 2024), that the LTI system using Hill's hyper-matrix from Eq. (18), with an accurate Fourier expansion, produces time series results identical to those of the LM. In terms of accuracy analysis for load cases with a stochastic input forcing, the exceedance probabilities for signals are extracted as the positive response peaks distance from the mean (steady-state) values.

We compare time- and frequency-domain results for the floater pitch angle $\xi_5$, as well as the first blade's ($l = 1$) deflection amplitude $a_1$ and dynamic stall separation function $f_{s,1}$. The response results for a single blade only suffice to describe the accuracy of the response calculation methods.

Prior studies using other frequency-domain solvers, such as RAFT, SLOW, and QuLAF which are mentioned in the Introduction, do not account for the azimuthal variation of linearized aerodynamic blade loads. Instead, they typically model the rotor load as concentrated at the hub. These studies often compare either frequency- or Laplace-domain results with time-domain simulations obtained from time-domain models, experiments, or higher-fidelity simulations (e.g. CFD or hydro- and aero-elastic solvers). However, they do not investigate different techniques for both frequency- and Laplace-domain simulations of azimuthally dependent linearized aerodynamic loads, nor do they assess the accuracy in comparison to both a linear and a more accurate time-domain model. Furthermore, the simplified floating wind turbine developed in this work offers a lower accuracy compared to more sophisticated aero-elastic solvers. Due to these differences in how aerodynamic and hydrodynamic loads are modeled, direct comparisons with previous models or experimental data are not feasible. The loading and structural modeling approaches used in previous studies differ significantly from the one presented here, limiting the applicability of benchmarking against high-fidelity numerical models or experimental results. Rather than serving as a benchmark tool, the simplified model introduced in this study is intended to explore differences between time-domain, linearized, and fast-response methods. Its streamlined formulation provides clear insights into system dynamics and method performance, complementing, rather than replacing, more detailed aero-elastic solvers such as HAWC2, Bladed, OpenFAST, or SIMA.

In this paper, five different load cases are analyzed where different inflow velocity and floater pitch moment $M_F$ are considered. The load cases distinctions are summarized in Table 1.



**Table 1.** Simulations load cases for the operational point of $V_0 = 8 \ \mathrm{m \, s^{-1}}$ and $\Omega = 0.6 \ \mathrm{rad \, s^{-1}}$

| Load Case | Aero: Inflow velocity | | | Hydro: Floater pitching moment | |
|:---:|:---:|:---:|:---:|:---:|:---:|
| | Constant | Sheared | Turbulent | Harmonic | Stochastic |
| A | ✓ | | | ✓ | |
| B | | ✓ | | ✓ | |
| C | | | ✓ | ✓ | |
| D | ✓ | | | | ✓ |
| E | | ✓ | ✓ | | ✓ |

The inflow velocity can either be constant, modified by a sheared perturbation, or influenced by coherent turbulence, which is stochastic in nature and characterized by a turbulence intensity (TI) of $5.77\%$. As for the floater pitch moment, it can be harmonic or stochastic. Among the load cases presented in Table 1, Case $E$ is considered the most realistic, as it accounts for a stochastic inflow velocity, a stochastic floater pitch moment, and a sheared inflow velocity profile.

## 6.1 Load case $A$: constant inflow and harmonic floater forcing

For the load case $A$ in Fig. 3, the inflow velocity is constant, and the time series indicate that the zeroth harmonic is insufficient to characterize accurately the blade responses compared to the full LM. A graphically good agreement, however, is seen for the first- and higher-order version of all the fast response methods and the Laplace-based method.

The log-scaled PSD plots in the blades DOFs, $a_1$ and $f_{s,1}$, indicate energy peaks at the frequencies distanced at $-\Omega$ (-1P) and $\Omega$ (1P) away from the floater excitation frequency $\Omega_M$. This demonstrates the frequency coupling caused by the periodic terms in the inertia matrix, also referred to as the mass matrix.

Regarding the floater pitch angle $\xi_5$, mainly $\Omega_M$ is influential on the floater pitch motion considering the high PSD peak occurring at that frequency. The DOF's own natural frequency is not noticeable at $\omega_{\xi_5}$ because the transient response is omitted in the PSD computation. For the blades DOFs PSDs, $a_1$ and $f_{s,1}$, the natural frequency $\omega_{\xi_5}$'s influence on the response is again not visible for the same aforementioned reason.

In a nutshell, the responses show energy peaks caused by the periodic inertia of the system. Due to the presence of a high aerodynamic damping, all results, irrespective of the load case, do not capture the blade's natural frequency $\omega_{1f}$.



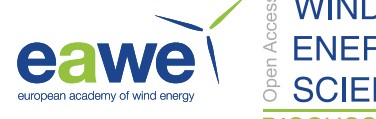

**Figure 3.** Time series and PSD plots for load case $A$ with the operational point of $V_0 = 8\,\mathrm{m\,s^{-1}}$ and $\Omega = 0.6\,\mathrm{rad\,s^{-1}}$.





## 6.2 Load case $B$: sheared inflow and harmonic floater forcing

The responses in Fig. 4 for load case $B$ are associated to a sheared inflow velocity and are highly periodic as indicated by
the PSD plots. The LM, fast response and the Laplace approaches all produce seemingly identical results, where higher-order
harmonic corrections do not appear to have any contribution. This implies that the accuracy of the methods is not improved
with higher harmonics response considerations for this load case.

The main frequency that is visible in the blade channels, $a_1$ and $f_{s,1}$, is $\Omega$ (1P), and the floater harmonic excitation frequency
$\Omega_M$ in the $\xi_5$ channel. Smaller PSD peaks are observable in the blade channels at the excitation frequencies $\Omega_M$ and $\Omega_M + \Omega$,
with only a barely discernible peak at $\Omega_M - \Omega$. The rotational speed frequency $\Omega$ is distinguishable in the time series channels
for the blade DOFs, due to the strong influence of the sheared inflow which creates a periodic aerodynamic load. The TDM
PSD plots for the blades DOFs, $a_l$ and $f_{s,l}$, capture additional peaks at other integers of the rotational speed, meaning at $2\Omega$
(2P) and $3\Omega$ (3P) for example. This demonstrates the higher harmonic coupling effects captured by the TDM but not by the
LM and other response methods due to the dominating effect of other frequencies related to the frequency $\Omega_M$.

For the floater pitch DOF $\xi_5$'s channel, like in load case $A$, mainly the floater pitch moment excitation frequency $\Omega_M$ is
captured in the PSD plot, and the sinusoidal motion of the floater is visible in the time series plot. Similarly to the load case
$A$ as well, the natural frequency $\omega_{\xi_5}$ is not apparent on any channel's PSD plot since the transient response is not taken into
account for the PSD calculation.



**Figure 4.** Time series and PSD plots for load case $B$ with the operational point of $V_0 = 8\,\mathrm{m\,s^{-1}}$ and $\Omega = 0.6\,\mathrm{rad\,s^{-1}}$.





### 6.3 Load case $C$: turbulent inflow and harmonic floater forcing

Concerning the load case $C$ results displayed in Fig. 5 that are generated for a spatially coherent turbulent inflow velocity, the time series indicate that there is no clear periodic response for the blade variables channels, $a_1$ and $f_{s,1}$. Overall, the PSD plots for this stochastic load case illustrate how broadbanded the response spectra are due to the effects of turbulence.

  For the $f_{s,1}$ channel, the time series show a small offset between the TDM and the other results, while no visible difference is observed between the LM, the Fourier-based fast response methods, and the Laplace-based method.

In the floater pitch angular motion channel, the natural frequency is recognizable at $\omega_{\xi_5}$ because it is excited by the stochastic load which dominates the response. Further, the energy at the floater pitch excitation frequency $\Omega_M$ is visible to a minor degree for the floater pitch motion channel.



**Figure 5.** Time series and PSD plots for load case $C$ with the operational point of $V_0 = 8\,\mathrm{m\,s^{-1}}$ and $\Omega = 0.6\,\mathrm{rad\,s^{-1}}$.





To analyze the accuracy of the various methods in detail, exceedance probability plots are presented in Fig. 6. In this paper, for exceedance probability plots, we evaluate the absolute relative difference at the maximum positive peaks that have the same

exceedance probability with respect to a reference value, being generally the LM value. The deviations are generally small between between the TDM and LM. The deviations error are obtained by a comparison of two peaks of the same exceedance probability value. The largest ones occur for the $f_{s,1}$ channel for the largest peaks with a difference going up to 14 %, whereas it goes up to 5 % for the $a_1$ channel. The mismatch of higher signal peaks for the fast response and Laplace-based results with the LM occurs at very low exceedance probabilities. This entails that overall results have a good agreement with the LM

and the error is small. The largest deviation from the LM is observed in the $a_1$ and $f_{s,1}$ channels for the fast response single perturbation method of accuracy going up to $O(\varepsilon^2)$, resulting in an error reaching 2.8 % for the largest peaks in both channels.

Exceedance probability results are sensitive to small deviations from the LM reference. Consequently, they show that the double perturbation method provides slightly more accurate results than the single perturbation. In addition, for the present load case, an increased harmonic order of consideration (up to perturbation $\varepsilon^2$ or $\delta^2$) does not indicate a considerable improvement

in accuracy.

Deviations from the LM also occur for the Laplace method in particular for the $\xi_5$ channel result.

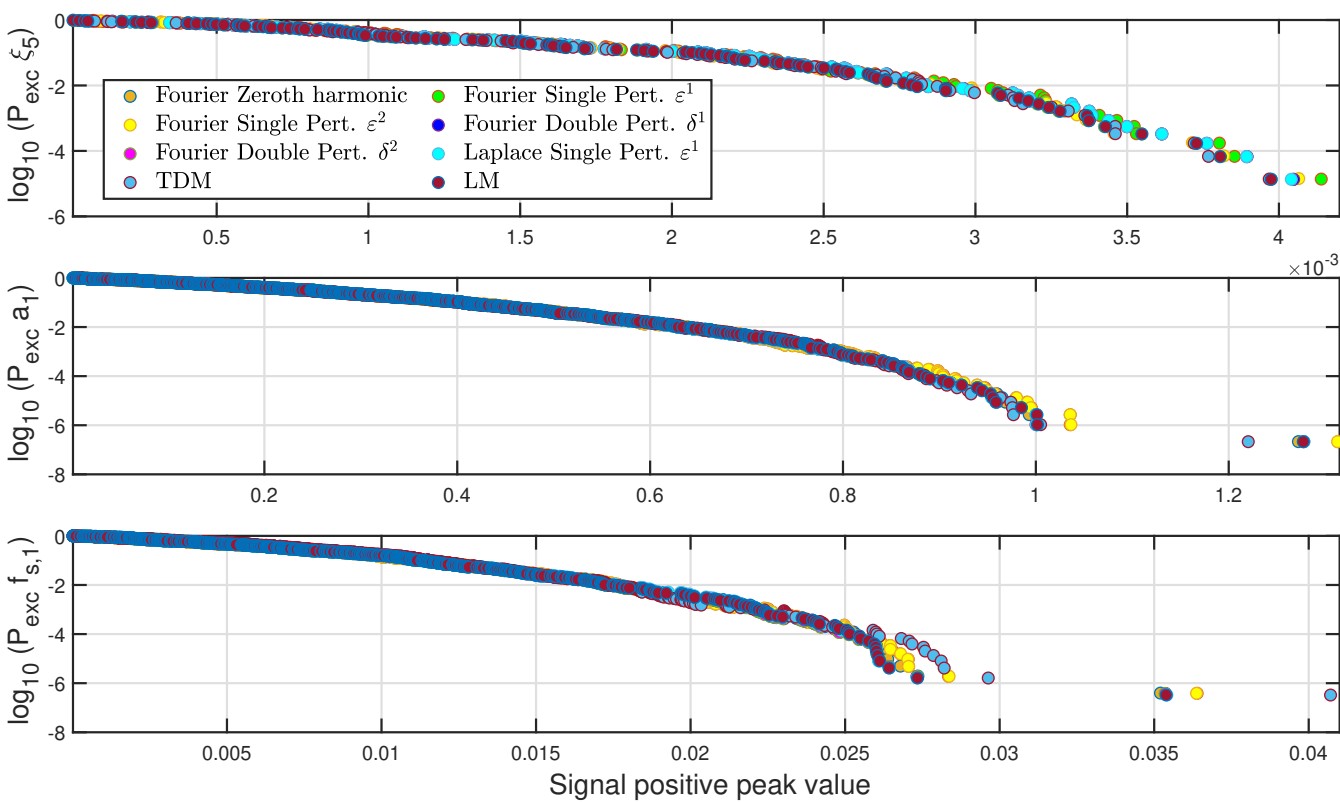

**Figure 6.** Logarithmic exceedance probability plots for load case $C$ evaluated for all the fast response methods, the time-domain model (TDM), and the linear model (LM). The operational point is $V_0 = 8 \, \text{m s}^{-1}$ and $\Omega = 0.6 \, \text{rad s}^{-1}$.





## 6.4 Load case $D$: constant inflow and stochastic floater forcing

For load case $D$ results shown in Fig. 7, only the floater pitch moment is stochastic. Generally, the PSD plots for this load case reveal a broadbanded response, as it is influenced by the stochastic nature of the floater pitch moment.

As observed in both the time series and PSD plots, the influence of the periodic system matrix at frequency $\Omega$ makes the zeroth harmonic alone insufficient to accurately represent the blade DOFs responses compared to methods that include higher harmonic effects. That being said, the comparison between the LM and the fast response and Laplace-based methods shows very good agreement when a higher-order harmonic accuracy is considered. There is also a small offset of the TDM response with the rest of results visible in the $f_{s,1}$ channel.



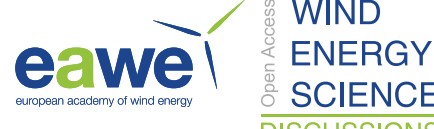

**Figure 7.** Time series and PSD plots for load case $D$ with the operational point of $V_0 = 8\,\mathrm{m\,s^{-1}}$ and $\Omega = 0.6\,\mathrm{rad\,s^{-1}}$.

 

In Fig. 8, as with the time series and PSD plots, the exceedance probability results for load case $D$ show some deviations between the LM and TDM results, with the largest deviation, an error of 8%, occurring in the $f_{s,1}$ channel. The zeroth harmonic results accurately approximate the floater pitch response but not the blade responses. Furthermore, the results that include contributions from at least one higher harmonic (fast response methods and the Laplace method) match the LM results perfectly for all channels.

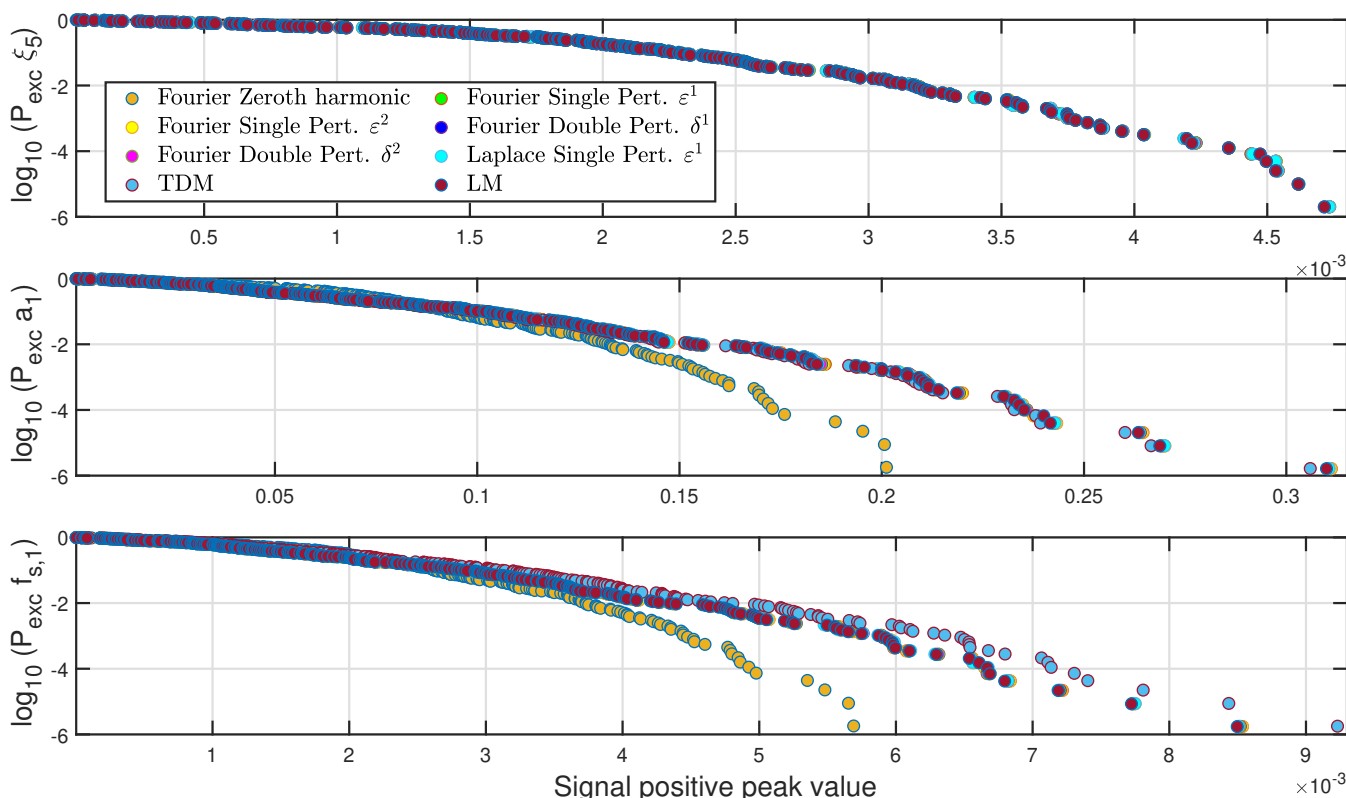

**Figure 8.** Logarithmic exceedance probability plots for load case $D$ evaluated for all the fast response methods, the time-domain model (TDM), and the linear model (LM). The operational point is $V_0 = 8\,\mathrm{m\,s^{-1}}$ and $\Omega = 0.6\,\mathrm{rad\,s^{-1}}$.

## 6.5 Load case $E$: shear, turbulent inflow and stochastic floater forcing

Finally, for the load case $E$, where both the inflow velocity and the floater pitch moment are stochastic, the results are showcased in Fig. 9. PSD plots for the stochastic load case $E$ reveal how broad-frequency the response is due to the influence of turbulent inflow.

The peak for the rotational speed frequency, $\Omega$, is noticeable in the floater pitch $\xi_5$ PSD channel but it is most apparent in the blade DOFs channels. Yet, it was also present on PSD plots for case $D$ but that peak was highly damped in comparison.

Generally, the results for the Laplace and fast response methods that are above the zeroth harmonic in accuracy match well with the LM.

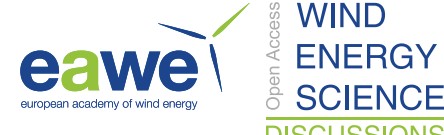

**Figure 9.** Time series and PSD plots for load case $E$ with the operational point of $V_0 = 8 \, \text{m s}^{-1}$ and $\Omega = 0.6 \, \text{rad s}^{-1}$.





The load case $E$ exceedance probability results are shown in Fig. 10. They overlap each other for the most part, except for the TDM results in the blade DOFs channels $a_1$ and $f_{s,1}$. Just like for the time series, the exceedance probability results for

the methods having an accuracy that is above the zeroth harmonic methods, agree well with the LM results. In that respect, the largest errors occurring in the $a_1$ and $f_{s,1}$ channels under load case $E$ reach a maximum of approximately 1.3%. There are additional deviations of the Laplace method from the LM results that appear in the $\xi_5$ channel and which are of 1.6 % error in magnitude.

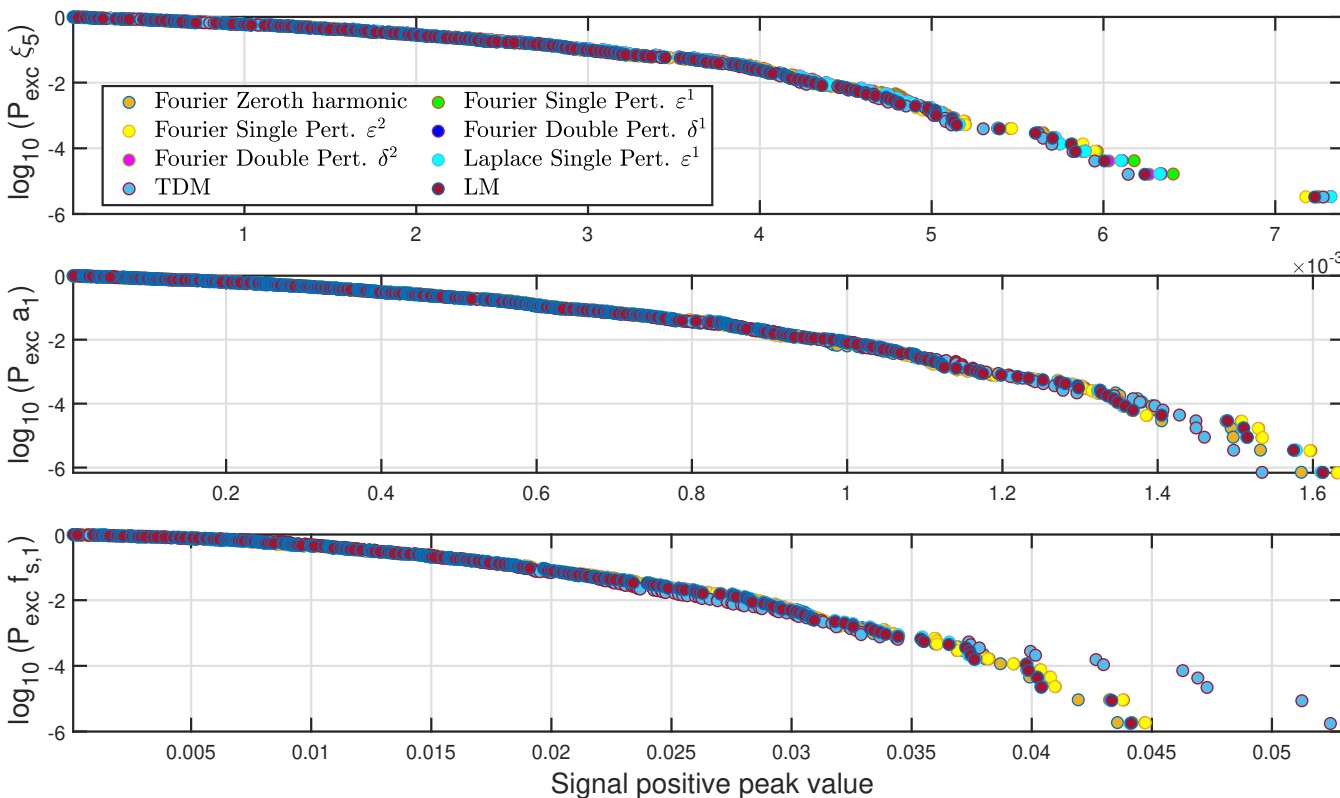

**Figure 10.** Logarithmic exceedance probability plots for load case $E$ evaluated for all the fast response methods, the time-domain model (TDM), and the linear model (LM). The operational point is $V_0 = 8 \, \mathrm{m\,s^{-1}}$ and $\Omega = 0.6 \, \mathrm{rad\,s^{-1}}$.

## 6.6 Overview

Overall, a noticeable discrepancy in results occurs between the TDM and LM. That is to be expected due to the non-linear effects that the TDM takes into account with the variability of aerodynamic variables. However, for most load cases, the results of the perturbation methods matched well with the LM reference. The deviations from the LM are not always perceptible in time series excerpts and PSD plots. They become noticeable in exceedance probability plots with an increasing signal peak value and a reduced probability.



An important mismatch is observed between the zeroth harmonic response and responses of a higher harmonic order consideration. This occurs when the forcing contains a high periodicity with $\Omega$ for a specific load case. This inaccuracy is visible in time series, PSD plots, and particularly in exceedance probability plots where deviations from the LM reference are most apparent.

To provide an overview of the accuracy of the results, the exceedance probability error is compared in terms of the signal
positive peak value relative to the LM at the level of $P_{\mathrm{exc}} = 10^{-2}$ and at the highest peak level, labeled $P_{\mathrm{exc}} = 10^{-x}$. If the relative error with respect to the LM is higher at another exceedance probability level than at the highest peak, which can sometimes occur (e.g. load case $E$), then the error is evaluated at that level. Besides, the relative error can be evaluated for the stochastic load cases $C$, $D$, and $E$, across the different response channels $\xi_5$, $a_1$, and $f_{s,1}$, and various response (Resp.) calculation methods. The response calculation methods include the Fourier single perturbation method with accuracy $\varepsilon^2$ (S2),
the double perturbation method with accuracy $\delta^2$ (D2), and the Laplace single perturbation method with accuracy $\varepsilon^1$ (L1). The relative error results for these three methods are presented in Table 2.

**Table 2.** Exceedance probability relative error in percentage (%) for signal positive peak value with respect to LM reference for the operational point of $V_0 = 8\,\mathrm{m\,s^{-1}}$, $\Omega = 0.6\,\mathrm{rad\,s^{-1}}$

| Load Case | Resp. | $P_{\mathbf{exc}} = 10^{-2}$ | | | $P_{\mathbf{exc}} = 10^{-x}$ | | |
|---|---|---|---|---|---|---|---|
| | | $\xi_5$ | $a_1$ | $f_{s,1}$ | $\xi_5, x$ | $a_1, x$ | $f_{s,1}, x$ |
| C | S2 | 2.239 | 1.546 | 0.868 | 2.282, 4.844 | 2.778, 6.670 | 2.777, 6.413 |
| | D2 | 0.127 | 0.091 | 0.118 | 0.002, 4.868 | 0.006, 6.670 | 0.110, 6.420 |
| | L1 | 1.750 | 0.146 | 0.846 | 1.670, 4.868 | 0.046, 6.666 | 0.063, 6.480 |
| D | S2 | 0.129 | 0.325 | 0.397 | 0.384, 5.694 | 0.361, 5.790 | 0.434, 5.759 |
| | D2 | 0.088 | 0.563 | 0.024 | 0.412, 5.694 | 0.108, 5.790 | 0.210, 5.759 |
| | L1 | 0.116 | 0.106 | 0.148 | 0.379, 5.694 | 0.198, 5.781 | 0.096, 5.753 |
| E | S2 | 0.588 | 0.479 | 0.725 | 1.599, 4.378 | 1.334, 5.468 | 1.338, 5.730 |
| | D2 | 0.084 | 0.122 | 0.206 | 0.434, 4.386 | 0.100, 5.462 | 0.085, 5.727 |
| | L1 | 0.224 | 0.017 | 0.266 | 1.607, 4.374 | 0.232, 5.451 | 0.109, 5.756 |

According to the relative error results in Table 2, the D2 method generally provides the highest response precision, while the L1 method occasionally outperforms it depending on the load case and the exceedance probability level considered. The overall accuracy of these two methods is excellent, with the highest observed relative error not exceeding 0.56%. In contrast,
the S2 method consistently yields the lowest accuracy across most load cases and exceedance probability levels, with relative errors systematically higher than those of the D2 and L1 methods. This discrepancy is particularly notable in the estimation of the $\xi_5$ channel response, where the S2 method often underperforms compared to its counterparts. The largest error for the S2 method is 2.78%, which is still fairly accurate.





## 6.7 Standard Deviation Relative Error

The accuracy of the fast response methods can be alternatively quantified through the Standard Deviation Relative Error (SDRE) which is denoted $\sigma_{SDRE}(\tilde{q}_{i,method}(t))$ and evaluated for each $i^{th}$ response channel's non-transient data samples. It is calculated through the standard deviation ($\sigma(\cdot)$) of the data samples' residual with respect to the LM reference values $\tilde{q}_{i,LM}(t)$, and then normalized with respect to the standard deviation of the LM values,

$$\sigma_{SDRE}(\tilde{q}_{i,method}(t)) = \frac{\sigma(\tilde{q}_{i,method}(t) - \tilde{q}_{i,LM}(t))}{\sigma(\tilde{q}_{i,LM}(t))}. \tag{33}$$

Evidently, a higher SDRE value translates to a lower accuracy. Compared to the analysis of exceedance probability plots in Figs. 6, 8 and 10, this error measure concerns a direct deterministic comparison of the response time series. The SDRE accuracy values are visualized for the fast response and Laplace methods in comparison to the LM benchmark in Fig. 11.

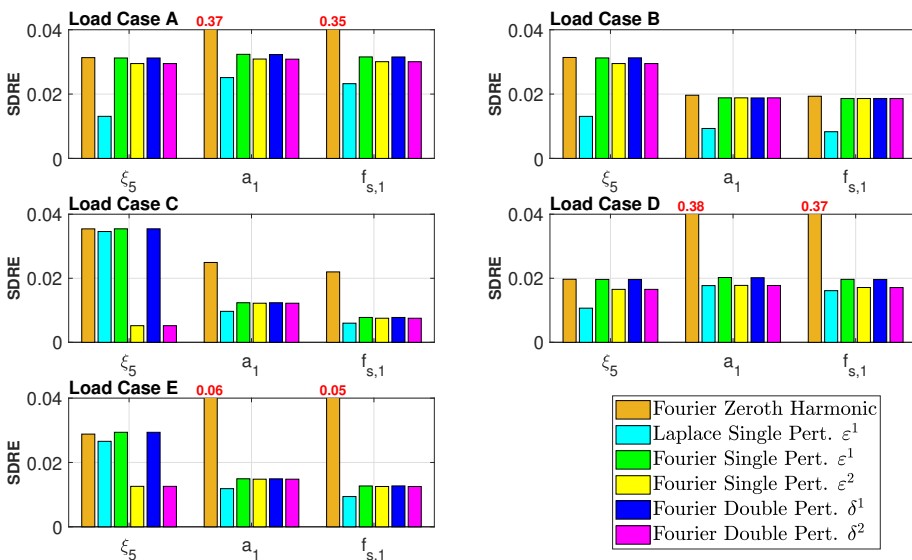

**Figure 11.** Standard Deviation Relative Error (SDRE) for varying load cases and response channels. The analyzed fast response methods are the Fourier zeroth harmonic, as well as the double and single perturbation (Pert.) methods.

Results in Fig. 11 confirm roughly the same observations as deduced from exceedance probability plots and time series. For the single and double perturbation methods, the accuracy was first tested for a precision up to first-order harmonic ($\varepsilon^1$ and $\delta^1$

perturbation). Then the response accuracy was increased up to second second-order harmonic ($\varepsilon^2$ and $\delta^2$ perturbation) which did improve it considerably for the $\xi_5$ channel in load cases $C$ and $E$, whereas it did not affect it significantly for other load cases. This supports the choice of settling for a maximal second-order harmonic accuracy being tested for both the single and double perturbation methods.

While the zeroth harmonic method can give error levels up to 38%, the error levels of both first-order methods (single and

double perturbation) is below 3.5 % for all tests. For some of the tests, the second-order methods improve the accuracy relative





to the first-order methods. Thus, e.g. in the stochastic load cases $C$, $D$ and $E$, they give error levels below 2%. As for the Laplace single perturbation method of $\varepsilon^1$ perturbation order, its accuracy fluctuates more than for fast responses but is below 3.5% for all load cases.

Moreover, there are some important numerical attributes of the system matrices worth noting that explain why part of the
results are not always affected by the load case itself in this study.

Firstly, the time-varying components of the system matrix $\tilde{\mathbf{A}}_L$ can be approximated by only the first-order harmonic contribution $\tilde{\mathbf{A}}_{L,1}$ meaning that other higher-order harmonics, including $\tilde{\mathbf{A}}_{L,2}$, are negligible, i.e. $\tilde{\mathbf{A}}_L \approx \tilde{\mathbf{A}}_{L,1}$ and $\tilde{\mathbf{A}}_{L,2} \approx \mathbf{0}$. Under this observation, the single and double perturbations numerical schemes produce identical accuracy results in terms of SDRE value. This implication is discernible through a comparison of the single and double perturbation Eqs. (28) and (24) respec-
tively. These equations are equal if we neglect the term $\tilde{\mathbf{A}}_{L,2}\tilde{\boldsymbol{q}}_0 \approx \mathbf{0}$ in Eq. (24). That explains why for most load cases the error levels do not decrease significantly from first- to second-order harmonic response and that the single and double perturbation methods show broadly identical accuracy.

Secondly, the off-diagonal terms in Eqs. (28) and (24) are forcing input contributions that multiply the transfer function $\mathbf{H}(\omega)$. When solving the first-order harmonic $\tilde{\boldsymbol{q}}_1$, the forcing contribution is only determined by $\tilde{\mathbf{A}}_L\tilde{\boldsymbol{q}}_0$ or $\tilde{\mathbf{A}}_{L,1}\tilde{\boldsymbol{q}}_0$ depending
on the method (single or double perturbation). For the $\xi_5$ channel in all load cases, this numerical forcing term happens to produce almost a null first harmonic response $\tilde{\boldsymbol{q}}_1 \approx \mathbf{0}$. In this scenario, the zeroth harmonic response, $\tilde{\boldsymbol{q}}_0$, magnitude (Euclidean norm) is much greater than the corresponding DOFs coefficients in $\tilde{\mathbf{A}}_{L,1}$ (Frobenius norm). This indicates that the coupling between these two terms of varying harmonic is indeed negligible, which is why there is no added accuracy in SDRE when adding the first-order harmonic contribution to the zeroth order for the $\xi_5$ channel.

One of the major distinctions between load cases accuracy is observable for the floater pitch channel $\xi_5$ when the load case considers the stochastic forcing from a turbulent inflow or a stochastic hydrodynamic moment. According to the corresponding PSDs for load cases $C$, $D$ and $E$ in Figs. 5, 7 and 9, there is a peak at the rotational speed $\Omega$ ($1P$). The peak at $1P$ frequency occurs within the power spectra in the $\xi_5$ channel for load case $C$, and in the blade DOFs channels, $a_1$ and $f_{s,1}$, for load cases $D$ and $E$ particularly. For these particular scenarios, the $\Omega$ frequency of excitation signifies that the system is influenced by the
corresponding first-order harmonic response $\tilde{\boldsymbol{q}}_1$. Given that $\tilde{\boldsymbol{q}}_1$ is non negligible for load cases $C$, $D$ and $E$, the evaluation of the forcing term $\tilde{\mathbf{A}}_{L,1}\tilde{\boldsymbol{q}}_1$ for the double perturbation method and $\tilde{\mathbf{A}}_L\tilde{\boldsymbol{q}}_1$ for the single perturbation method, improves the model accuracy through the contribution of a second-order harmonic $\tilde{\boldsymbol{q}}_2$ in the response $\boldsymbol{q}$. This improvement in accuracy translates to a decrease in SDRE for the $\xi_5$ channels.

The SDRE has also been evaluated individually for load case $C$, considering variations in the stochastic inflow turbulence
intensity (TI), and for load case $D$, by repeating simulation runs with different seeds for the stochastic hydrodynamic moment. The findings for the variation of TI in load case $C$ and the different runs in load case $D$ are illustrated in Appendices C1 and C2, respectively. According to the corresponding two Figs. C1 and C2, there is visibly a stronger variation of SDRE values for the load case $C$ with changes in the inflow TI, compared to the load case $D$ with different simulation runs and different inputs of stochastic hydrodynamic moments. For load case $D$, changes in the stochastic seed have minimal impact on the bar plots,
while TI variations for load case $C$ have a slightly more pronounced effect on the SDRE results.





Starting from the analysis of the SDRE for all load cases $A$ to $E$ in Fig. 11, it is clear that the variability in the SDRE values is not due to random noise but reflects a method-dependent sensitivity to different stochastic excitations. To rigorously assess these differences for all response methods, non-parametric statistical tests were applied. They are more suitable than traditional parametric approaches, such as the analysis of variance (ANOVA) or t-tests which presume normality and equal

variances of responses. Non-parametric tests were performed due to the non-Gaussian and nonlinear nature of the system responses under turbulent and stochastic hydrodynamic moment inputs. In particular, the Kruskal-Wallis test was used for independent sample comparisons in order to treat responses from each method as unrelated, while the Friedman test accounted for repeated response measures when the same system was analyzed using different estimation methods. The Kruskal-Wallis statistical analysis produced $p$-values (probability values) well below the 0.05 threshold for load cases $C$ (turbulent inflow) and

$D$ (stochastic hydrodynamic loading), and for all three output channels $\xi_5$, $a_1$, and $f_{s,1}$. This indicates statistically significant differences between the estimation methods. When all load cases ($A$–$E$) are considered, the $p$-values increased slightly above 0.05, suggesting that the influence of the method becomes more diffuse across a broader range of conditions. This observation is logical, as the influence of the method on the SDRE value is more pronounced across varying load cases $A$ to $E$ than across the variation of turbulence intensity (TI) within load case $C$. In that vein, the lowest $p$-values are observed for load case $D$, as

the variation between simulation runs has less impact on the SDRE compared to the more substantial changes introduced in the other load case groupings. Friedman test results revealed that accross all load conditions, including the combined set $A$ to $E$, $p$-values remained consistently below 0.01 for all three response channels. This suggested a stronger and more systematic impact of the estimation method when repeated measures from the same simulations were considered.

    While both the Kruskal–Wallis and Friedman tests indicated that at least one method differs from the others, they did

not reveal which ones. Therefore, post-hoc pairwise comparisons were conducted to identify the specific method pairs with statistically significant differences. Hence, the post-hoc tests revealed for each channel and load case scenario, which pairs of methods differ the most from each other and which method stands out from other methods. In general, the zeroth harmonic and Laplace methods consistently showed statistically significant differences compared to other methods across most load case scenarios and response channels. This is to be expected, since the single and double perturbation methods follow a similar

computational approach.

## 7   Fast response methods efficiency analysis

We now evaluate the execution time for various durations of the simulated time series. Such a comparison involves several choices that can significantly affect the results, including the simulation time step interval, $dt$, the load case being simulated, and the structure of the code. For instance, the simulation time step increment was set to $dt = 0.0937$ s which corresponds

to a Nyquist frequency of 5 Hz. The computational efficiency study was only carried out for load case $E$, because it is more realistic due to its stochastic nature both in wind inflow and hydrodynamic moment.

    Efficient implementations were assembled for both the single and double perturbation methods by avoiding looping over time steps when possible. Looping over time steps is, however, unavoidable for the Laplace single perturbation method approach





and when generating the system forcing time series $\boldsymbol{F}_{B,L}(t)$. But the inverted structural mass matrix $\mathbf{M}_S^{-1}(t)$ which affects

$\boldsymbol{F}_{B,L}(t)$ can be pre-computed for all time steps.

Since the new response methods require some pre-processing, CPU time can be saved by storing these results. This is outlined for the different response calculation methods in Appendix D, distinguishing between the first simulation for one random seed (Seed 1) and the subsequent simulation of a new seed (Seed 2). The hardware configuration used for computations was an HP EliteBook 840 G8 Notebook PC that features an Intel Core i7-1185G7 processor with 4 cores, 8 threads, a base

clock of 2.99 GHz, with up to 4.8 GHz turbo boost. Regarding the programming implementation in MATLAB of the response calculations there was no parallelization strategy. Not using parallelization in fast response analysis programming can be advantageous because it minimizes overhead costs, decreases latency, and enhances predictability. This simpler approach also improves cache efficiency and eliminates the complexities and synchronization delays that are often encountered in parallel processing. If more cores are available, several load cases can be run independently in parallel. The CPU time of the TDM and

LM for load case $E$ is shown in Fig. 12 with a log-log scale on the $x$- and $y$-axis. Fig. 12 also presents CPU time results for a Seed 1 for the fast response methods and a Seed 2 as well for the Laplace method.

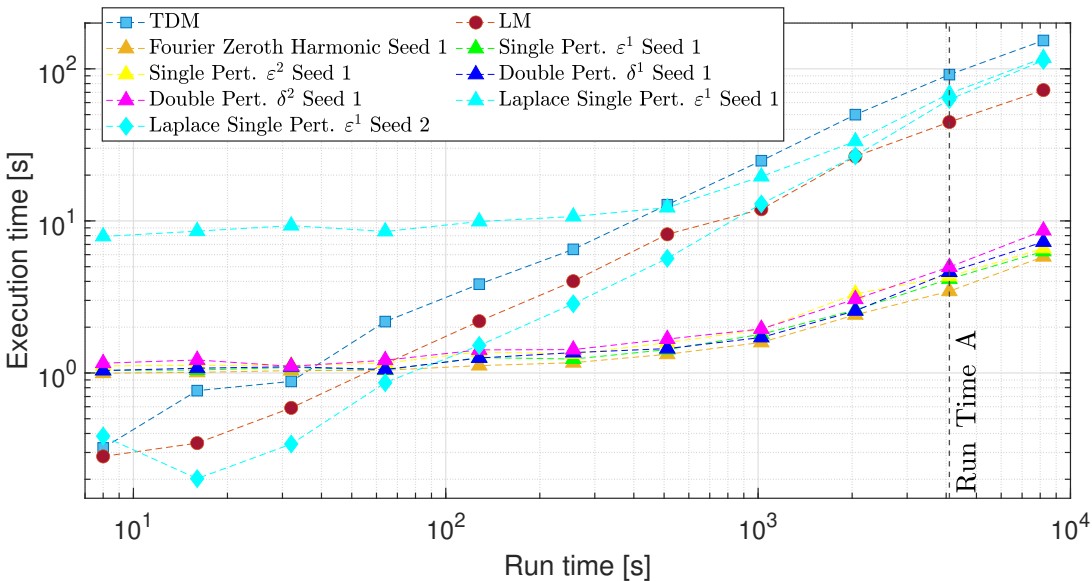

**Figure 12.** CPU execution time with respect to run time. Results are using the Seed 1 scheme for the Fourier-based fast response methods and the Seed 2 scheme as well for the Laplace method.

For the TDM, LM and the Laplace method with the Seed 2, the CPU time is proportional to time simulated with a slope that is very close to 1 in a log-log scale. In contrast, the CPU time for fast response methods has a plateau trend until reaching about 500 s of simulated time. Onward from that point, the fast response methods curves becomes straight with almost a unit slope

and then the CPU time is about 10 times smaller for a first seed simulation than for the LM. Similarly, for the Laplace method with a Seed 1, the CPU time has a plateau value until reaching 1000 s run time, and then onward it has a slope close to 1.

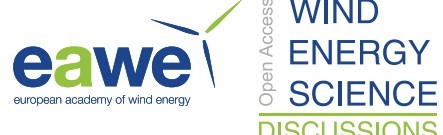

Compared to fast response methods, that run time threshold is highest obviously for the Laplace method with Seed 1 due to its higher computational cost. Before that run time is reached, the Laplace method with a Seed 1 is more computationally expensive in execution time than any other method including the LM and TDM. Also, the Laplace method using the Seed 1 scheme is at all run times less efficient than the LM. Due to the time step looping procedure, the Laplace method is computationally vastly more costly than all other Fourier-based methods. The Laplace method using the Seed 2 scheme is slightly less efficient than the LM at higher run times, otherwise the efficiency is considerably close to the LM results and varies with the same trend. The Laplace single perturbation method is the slowest of the methods we developed, mainly because of the time-looping process used to solve the response and the preliminary symbolic operations needed. A single time-loop was tested to solve both the zeroth harmonic $q_0$ and first harmonic $q_1$ responses, but it was found to be slower than using two separate loops to solve each response individually. Additionally, the Fourier-based fast response methods with a Seed 1 are only less efficient than the LM until reaching a run time of $60$ s but after exceeding that run time threshold they become more efficient than all other methods.

The CPU time required for a Seed 2 computation using the fast response methods for load case $E$ is shown in Fig. 13, with both the $x$- and $y$-axes displayed on a log-log scale. There is now a one to one ratio between the execution and run time because there are no initialization costs for simulations carried out with a Seed 2. For example, with a run time of 4096 s (Run Time $A$ on Figs. 12 and 13), the execution time for fast response methods using a single or double perturbation approach, such as the single perturbation approach with a response correction up to the second-order ($\varepsilon^2$), is 1638 times faster than the real run time. It is also 36 times faster than the TDM, and 18 times faster than the LM. Since the inverted structural mass matrix can be stored and other operations can be pre-computed for a Seed 1 simulation, the execution time is further reduced by a factor 2 for a Seed 2 simulation.

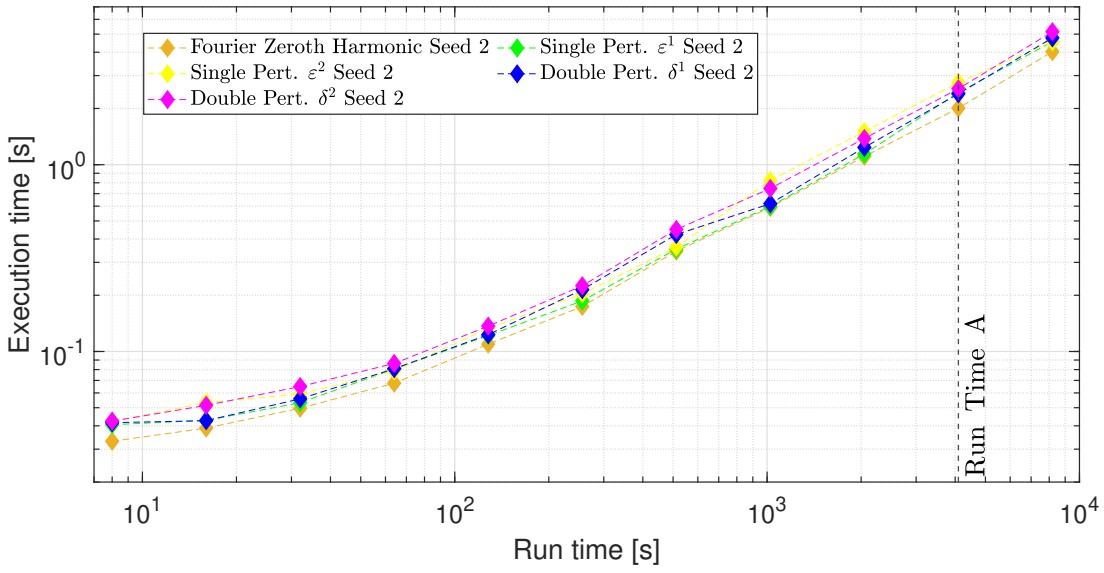

**Figure 13.** CPU execution time for Seed 2 scheme with respect to run time.





The results for Seeds 1 and 2, in Figs. 12 and 13, point out as anticipated that for both the single and double perturbation results the single harmonic computation (up to perturbation $\varepsilon^1$ or $\delta^1$) requires less execution time than up to the second harmonic (up to perturbation $\varepsilon^2$ or $\delta^2$).

In the end, the fastest fast response FFT method is the zeroth harmonic response contribution only. Due to a lack of accuracy, we find that the best alternative method is the higher-order single perturbation method with a second-order response correction. It is generally faster than the double perturbation method and provides a quasi-identical accuracy. Depending on the load case, a second-order consideration did improve the accuracy as shown in Fig. 11 and the additional computational cost for adding the second-order correction is small. As an example of accuracy evaluation, one can take into account results again for the run time of 4096 s (Run Time $A$). For this scenario, the first-order (up to first harmonic precision) single perturbation method predicts

a result with an accuracy of 3.5% (Fig. 11) within 2.5 seconds CPU time. The second-order single perturbation method further improves that accuracy substantially for some load cases as can be seen in Fig. 11.

Among the various performance metrics analyzed, one particularly informative aspect is the overhead or initialization cost incurred by different computational schemes. The results presented in Fig. 14 illustrate the overhead costs that apply when performing simulations according to the Seed 1 computational scheme. These findings complement earlier results by highlighting

more the trade-offs between accuracy and computational efficiency.

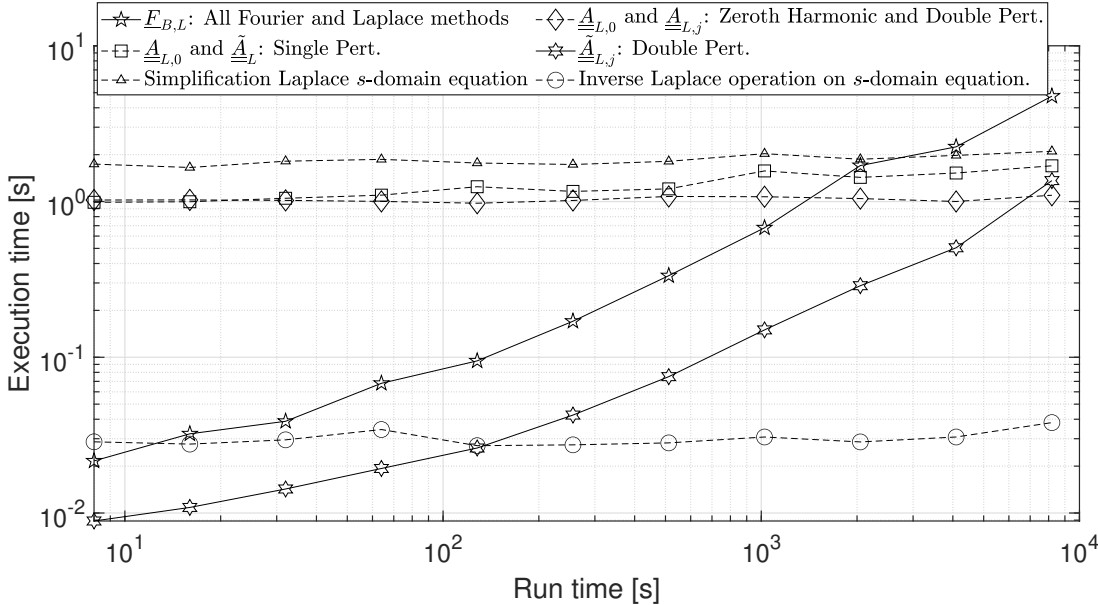

**Figure 14.** Fast response methods overhead costs in terms of execution time versus run time. The methods which require each operation are specified.

For the Laplace method, the highest initialization cost contribution comes from the symbolic simplification procedure of the Laplace $s$-domain Eq. (31), rather than from solving it symbolically.





Another trend that is noticeable in Fig. 14 is that both the forcing time series generation $\boldsymbol{F}_{B,L}(t)$ and the generation of the double perturbation numerical input forcing term $\tilde{\mathbf{A}}_{L,j}(t)$ are procedures that increase with a unit slope in execution duration
with simulation run time. While the total computational cost of $\mathbf{A}_{L,0}$ and $\tilde{\mathbf{A}}_L$ in the single perturbation approach increases with the number of time steps, the growth rate is significantly lower than for the computation of $\tilde{\mathbf{A}}_{L,j}(t)$, owing to the faster evaluation of the matrix $\tilde{\mathbf{A}}_L(t) = \mathbf{A}_L(t) - \tilde{\mathbf{A}}_{L,0}$ that grows in size. As expected, the cumulative computation of the matrix $\mathbf{A}_{L,0}$ and higher harmonic matrices $\mathbf{A}_{L,j}$ via Hill's decomposition remains unaffected by the difference in run time, and it is required for both the zeroth harmonic and the double perturbation methods. Lastly, the Laplace method operations also remain
theoretically constant with runtime variations, however fluctuations are observed due to the stochastic nature of the runs.

We also evaluated the efficiency of our implementation by measuring the proportion of execution time spent on inverting the structural mass matrix $\mathbf{M}_S$ during the generation of the forcing time series $\boldsymbol{F}_{B,L}(t)$ across all time steps. Our observations showed that it reaches a maximum level of about 10 %. With that plateau value reached as the run time increases, we can consider that only a small proportion of the forcing time series generation $\boldsymbol{F}_{B,L}(t)$ (Eq. (15)) is spent on that operation.
This means that this procedure does not require optimization and can remain unchanged without the need to further reduce computational costs.

## 8 Conclusions

We have developed novel methods in both the frequency and Laplace domains to enable rapid analysis of the aero-elastic behavior of floating wind turbines. The proposed Fourier-based and Laplace-based perturbation techniques model the system's
dynamic response with harmonic accuracy up to second order. They are intended to serve as practical tools for early-stage design and optimization, providing accurate insights into system behavior while significantly reducing the computational cost compared to traditional time-domain simulations.

These computationally efficient methods were validated under various load cases, including constant inflow, sheared inflow, and stochastic waves and wind. By introduction of a harmonic ordering parameter, the original linear model with a time-varying
system matrix was reformulated into a sequence of linear problems with a constant system matrix, suited for solution in the frequency domain. These new methods vary first based on the perturbation approach which can either be single or double. They can both be evaluated up to a chosen harmonic order. Hereby, higher-order harmonic corrections were shown to improve accuracy at minimal computational cost. Results demonstrated that zeroth-order approximations were insufficient in cases of strong periodicity, whereas the inclusion of higher harmonics significantly improved fidelity. Further, the Laplace single
perturbation method is the slowest among the methods we developed, primarily due to the time-looping procedure involved in solving the response and the symbolic operations required beforehand.

In terms of computational performance, after initialization costs, the CPU time of the fast response methods scaled proportionally to the simulation time, and achieved speedups of 8 to 10 times relative to the linear baseline. The response accuracy is adjustable via the number of harmonic corrections considered. In the end, an accuracy going up to the second-order perturba-
tion was sufficient to obtain reliable results for all load cases that were studied. The second-order single perturbation offered





the best trade-off between speed and accuracy, with an improved speed compared to the double perturbation method. With the linear model used as reference, it achieved a standard deviation relative error below 3.5% and an error below 3% with largest positive peaks comparison. Conversely, for the second-order double perturbation method, the error level for the largest response peaks across all load cases and output channels was below 0.5%. Although the second-order double perturbation method generated a slightly higher accuracy, it required approximately 25% more CPU time.

The numerical methods that we elaborated are based on several assumptions and possess few limitations. While the proposed approaches assume a linear system behavior, which limits the capture of nonlinear and transient effects, such assumptions remain appropriate for preliminary design analyses. The difference between the time-domain model and the linear model was quantified to assess the accuracy of the linear approximation. For few load cases, a small difference was observed between the time-domain and linear model results across the time series, PSD, and logarithmic exceedance probability plots, as the time-domain model accounts for nonlinear effects resulting from variations in aerodynamic parameters. Additionally, even though frequency-domain analyses are limited in their ability to comprehensively represent transient and nonlinear effects, to address this issue, we devised a Laplace domain method that considers transient response effects from initial conditions and relies on a first-order harmonic approximation. Consequently, it captured accurately the system's dynamic characteristics across most load cases. Furthermore, the use of the Øye dynamic stall model was beneficial for computational efficiency and straightforward implementation. Although the Øye dynamic stall model can be extended to incorporate shed vorticity effects that are accounted for in the Beddoes-Leishman model via Theodorsen's function, it was used here in its original form for the sake of clarity and demonstration.

Finally, the developed response calculation methods are compatible with state-of-the-art time-domain solvers such as Bladed, OpenFAST and HAWC2. Thus they provide a means to obtain model-consistent fast linearized simulation results for use in load cases screening, pre-design optimization, and control development.

*Code availability.* The MATLAB code used for simulations and the numerical data are provided upon request to the main author.

*Author contributions.* BP was primarily responsible for writing the paper and independently developed the entire programming framework used for the simulations. He played a key role in the conceptualization of the models, the exploration of methods, and the generation, validation, and visualization of results. HB, TK and WY contributed to the development of the methodology, as well as to the conceptual and investigative aspects of the work. They actively participated in the revisions and editing process.

*Competing interests.* The corresponding author declares that none of the authors have any competing interests.





## Appendix A: Velocity triangle

To calculate the aerodynamic load $F_{l,aero}$, w e quantify the relative velocity $V_{rel,l}$ by analyzing the velocity triangle in Fig.
A1.

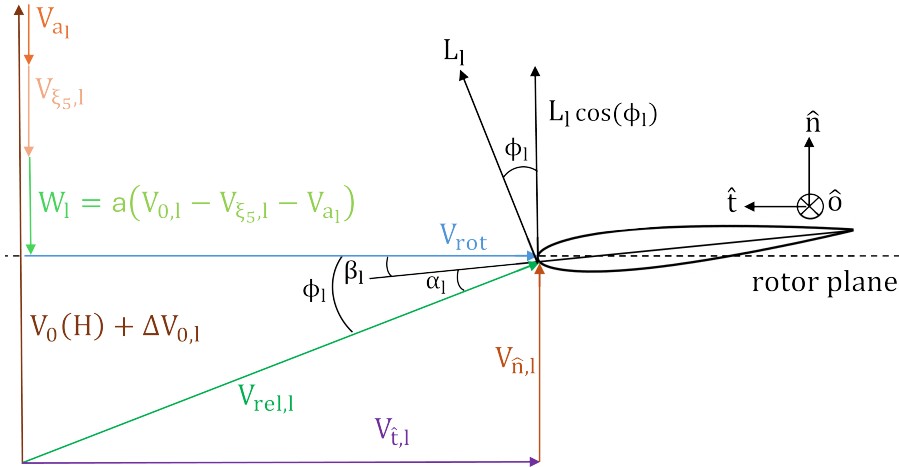

**Figure A1.** Velocity triangle for the airfoil located at the radial position of $r = d$ on the blade.

The integrated normal force $F_{l,aero}$ on the $l^{th}$ blade is given by

$$F_{l,aero} = \underbrace{L_l \cos \phi_l}_{F_l} L_b = \frac{1}{2}\rho c\, C_{L,l} V_{rel,l}^2 \cos \phi_l L_b, \tag{A1}$$

where $\phi_l$ is the inflow angle evaluated for the airfoil at $r = d$. In this study, it is assumed that the drag and tangential induced velocity contributions are relatively small compared to the lift and axial induction respectively, and can thus be neglected. While
assuming additionally the same inflow angle $\phi_l$ for the entire blade, we ignore the effects of variation in blade deformation across the blade. In Eq. (A1), for linearization purposes, $F_l = L_l \cos \phi_l$ represents the aerodynamic load contribution that dictates the floating wind turbine's dynamics.

As illustrated in Fig. A1, the relative velocity $V_{rel,l}$ component that is normal to the rotor plane, $V_{\hat{n},l}$, is impacted by a steady-state constant wake induction factor $a$. Conversely, the velocity tangential to the rotor plane, $V_{\hat{t},l}$, is approximated not
to be impacted by a tangential induction wake factor, $a'$, since this is usually small. Therefore, $V_{\hat{t},l}$ is equal to the rotational speed itself $V_{rot} = -\Omega d$. Additional relations are found with respect to the inflow angle $\phi_l$, the angle of attack $\alpha_l$ and the twist angle $\beta_l$, such as $\phi_l = \alpha_l + \beta_l$, and $\phi_l = \tan^{-1}\left(-V_{\hat{n},l}/V_{\hat{t},l}\right)$. The velocity normal to the rotor plane is given by $V_{\hat{n},l} = (1-a)\left(V_{0,l} - V_{\xi_5,l} - V_{a_l}\right)$, where $V_{0,l} = V_0(H) + \Delta V_{0,l}$, $V_{\xi_5,l} = \dot{\xi}_5\left(H + d\cos\Psi_l\right)$ and $V_{a_l} = \dot{a}_l \phi_{1f}(d)$. This entails that the





squared normal velocity $V_{\hat{n},l}^2$ can be expanded as

$$
\begin{aligned}
V_{\hat{n},l}^2 = (1-a)^2 \Bigg( &\underbrace{V_0^2(H)}_{\text{steady term}} + \underbrace{2\dot{\xi}_5\left(H+d\cos\Psi_l\right)\dot{a}_l\phi_{1f}(d)}_{\text{higher-order term neglected}} \\
&+ \underbrace{\dot{\xi}_5^2\left(H+d\cos\Psi_l\right)^2 + \dot{a}_l^2\phi_{1f}(d)^2}_{\text{higher-order terms neglected}} \\
&+ \underbrace{2\left(V_0(H)+\Delta V_{0,l}\right)\left(-\dot{\xi}_5\left(H+d\cos\Psi_l\right)-\dot{a}_l\phi_{1f}(d)\right)}_{\text{damping contribution}} \\
&+ \underbrace{\Delta V_{0,l}^2 + 2V_0(H)\Delta V_{0,l}}_{\text{forcing contribution}} \Bigg),
\end{aligned}
\tag{A2}
$$

where $\Delta V_{0,l} = \Delta V_{0,l,shear} + \Delta V_{0,turb}$. The damping contribution of the shear inflow velocity variation $\Delta V_{0,l,shear}$ is considered within the system structural damping matrix $\mathbf{C}_S$. This applies because of the periodicity of $\Delta V_{0,l,shear}$ according to Eq. (7). From Eq. (A2), we neglect however the contribution of the term $2\Delta V_{0,turb}\left(-\dot{\xi}_5\left(H+d\cos\Psi_l\right)-\dot{a}_l\phi_{1f}(d)\right)$ which is influenced by the spatially coherent velocity fluctuation $\Delta V_{0,turb}$.

Moreover, partial derivatives involving the inflow angle $\phi_l$, i.e. $\frac{\partial\phi_l}{\partial\cdot}$ and $\frac{\partial cos\phi_l}{\partial\cdot}$, become relevant when linearizing the system equations. They are related to the derivation of the normal velocity $V_{\hat{n},l}$ as shown in our previous paper's Eqs. (24) and (25) (Pamfil et al., 2025).

## Appendix B: Øye dynamic stall model and related studies

The Øye dynamic stall model comprises of an equation for the dynamic lift coefficient $C_L$ and for the separation function $f_s$ that influences its behavior. This model is linearized by applying Eq. (8) first for the dynamic lift coefficient $C_L$ from Øye's model Øye (1991),

$$
C_{L,l}\left(\alpha_l,f_{s,l}\right) = f_{s,l}C_{L,inv}\left(\alpha_l\right) + \left(1-f_{s,l}\right)C_{L,stall}\left(\alpha_l\right).
\tag{B1}
$$

In our anterior studies using this dynamic stall model (Pamfil et al., 2025), we have described in Eq. (29) the linearized terms $\frac{\partial C_{L,l}}{\partial\alpha_l}$ and $\frac{\partial C_{L,l}}{\partial f_{s,l}}$. Using the airfoil data from Fig. 3 in our investigations (Pamfil et al., 2025), the values of $\left.\frac{\partial C_{L,inv,l}}{\partial\alpha_l}\right|_{\text{st}}$ and $\left.\frac{\partial C_{L,stall,l}}{\partial\alpha_l}\right|_{\text{st}}$ are numerically evaluated under steady-state conditions ($st$) as gradients at the operating angle of attack $\alpha_l$ through the cubic spline interpolation. To quantify $\frac{\partial C_{L,l}}{\partial\alpha_l}$, the relation between the angle of attack and inflow angle must be considered according to Eq. (30) (Pamfil et al., 2025). At last, the Ordinary Differential Equation (ODE) for the dynamic stall





separation function $f_{s,l}$, $\dot{f}_{s,l} = \left( f_{s,static,l} - f_{s,l} \right) / \tau$, is linearized as

$$
\begin{aligned}
\dot{f}_{s,l,lin} = & -\frac{f_{s,l}}{\tau} + \\
& \frac{1}{\tau} \left( f_{s,static}\big|_{st} + \frac{\partial f_{s,static,l}}{\partial \alpha_l}\bigg|_{st} \frac{\partial \phi_l}{\partial \dot{\xi}_5}\bigg|_{st} \dot{\xi}_5 + \right. \\
& \left. + \frac{\partial f_{s,static,l}}{\partial \alpha_l}\bigg|_{st} \frac{\partial \phi_l}{\partial \dot{a}_l}\bigg|_{st} \dot{a}_l + \frac{\partial f_{s,static,l}}{\partial \alpha_l}\bigg|_{st} \frac{\partial \phi_l}{\partial \Delta V_{0,l}}\bigg|_{st} \Delta V_{0,l} \right),
\end{aligned} \tag{B2}
$$

where the time constant is $\tau = (4c)/V_{rel,st}$. The partial derivative $\frac{\partial f_{s,static,l}}{\partial \alpha_l}\big|_{st}$ required for the LM is obtained by numerically computing the gradient at the relevant operating angle of attack $\alpha_l$, based on the airfoil data presented in Fig. 4 from our published paper (Pamfil et al., 2025). As a reminder, the dynamic stall model, like the floating wind turbine model, is employed solely as a demonstration platform to validate the fast response methodology, and is not intended to constitute a central contribution of this study. Øye's dynamic stall model has only a single state and does not account for the separation effect of the vorticity that is shed from the airfoil's trailing edge. This phenomenon is captured analytically by Theodorsen's function and it is incorporated in the Beddoes-Leishman dynamic stall model (Leishman and Beddoes, 1986). However, we selected the Øye model for this analysis because it simplifies both the implementation and the linearization of the dynamic stall equations within the state-space framework. The validity of the linearization of the Øye dynamic stall model has been verified using the same dynamic model as in our previous published work (Pamfil et al., 2025). Therefore, in Fig. 6 (Pamfil et al., 2025) we compared for both the LM and the TDM the dynamic lift and stall behavior with a periodic floater pitch $M_F$ excitation. The analysis carried out at three different operational points near the region of maximum static lift coefficient $C_{L,static}$ demonstrated that overall, there is a good agreement between the TDM and LM time series for the angle of attack $\alpha$ and the lift coefficient $C_L$. A more extensive dynamic stall analysis could compare the behavior of multiple dynamic stall models. In pursuit of this goal, DNV developed a dynamic stall state-space model within its aeroelastic code Bladed, named IAG (Bangga et al., 2023), in reference to the Institute of Aerodynamics and Gas Dynamics at the University of Stuttgart, reflecting the developer's former affiliation. Their study investigated how the Øye, Beddoes–Leishman, and IAG dynamic stall models respond to varying excitation frequencies in edgewise vibrations of large and flexible wind turbine blades (Bangga et al., 2023), offering valuable insights for blade design. However, choosing the appropriate dynamic stall model is left to the reader, depending on the specific operating conditions, and conducting such a sensitivity study is beyond the scope of this work.





## Appendix C: Supplementary statistical analysis

### C1 Load case $C$: variation of inflow turbulence intensity

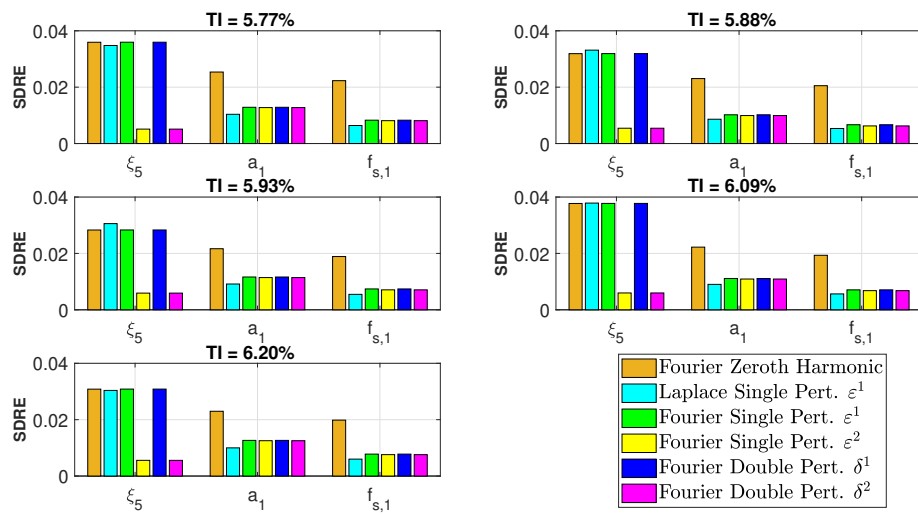

**Figure C1.** Standard Deviation Relative Error (SDRE) for load case $C$ turbulence intensity (TI) variations and response channels. The analyzed fast response methods are the Fourier zeroth harmonic, as well as the double and single perturbation (Pert.) methods.

### C2 Load case $D$: variation in stochastic hydrodynamic moment with multiple simulation runs

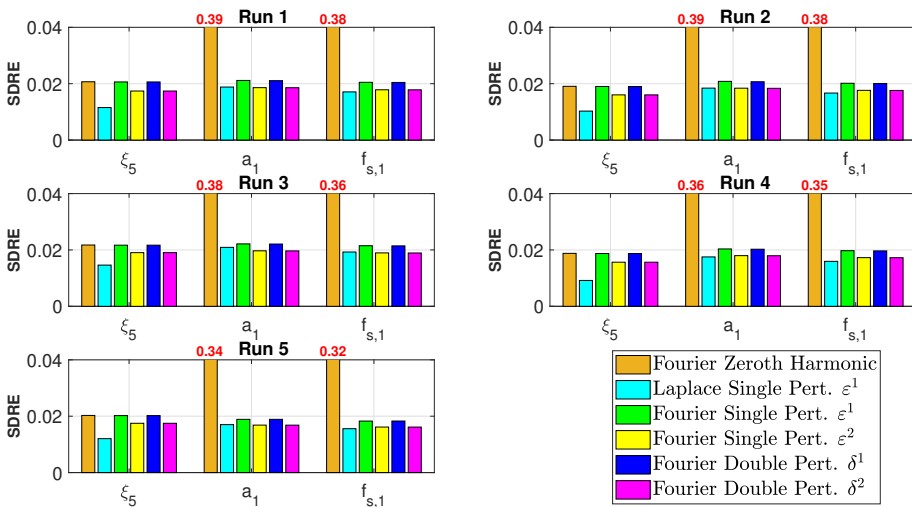

**Figure C2.** Standard Deviation Relative Error (SDRE) for load case $D$ multiple runs with different stochastic floater pitch moment and response channels. The analyzed fast response methods are the Fourier zeroth harmonic, as well as the double and single perturbation (Pert.) methods.





**Appendix D: Simulation protocols for Seeds 1 and 2**

The simulations protocols are summarized with the following steps for the zeroth harmonic and higher-order perturbation
(Pert.) methods:

**Appendix: Zeroth Harmonic**

    Seed 1:  (a) Compute $\boldsymbol{F}_{B,L}$ and store $\mathbf{M}_S^{-1}$ (Eq. (15)).

           (b) Calculate $\mathbf{A}_{L,0}$ via Hill decomposition (Eq. (21)).

    Seed 2:  (a) Compute $\boldsymbol{F}_{B,L}$ using stored $\mathbf{M}_S^{-1}$.

(b) Use stored $\mathbf{A}_{L,0}$.

**Appendix: Single Pert.**

    Seed 1:  (a) Compute $\boldsymbol{F}_{B,L}$ and store $\mathbf{M}_S^{-1}$.

           (b) Calculate average $\mathbf{A}_{L,0}$ and higher first-order harmonic $\tilde{\mathbf{A}}_L = \mathbf{A}_L - \mathbf{A}_{L,0}$ for time series (Eq. (21)).

    Seed 2:  (a) Compute $\boldsymbol{F}_{B,L}$ using stored $\mathbf{M}_S^{-1}$.

(b) Use stored $\mathbf{A}_{L,0}$ and $\tilde{\mathbf{A}}_L$.

**Appendix: Double Pert.**

    Seed 1:  (a) Compute $\boldsymbol{F}_{B,L}$ and store $\mathbf{M}_S^{-1}$.

           (b) Calculate $\mathbf{A}_{L,0}$ and harmonics $\mathbf{A}_{L,j}$ ($j > 0$) via Hill decomposition (Eq. (21)).

           (c) Compute $\tilde{\mathbf{A}}_{L,j}$ for time series using $\mathbf{A}_{L,j}$ (Eq. (21)).

Seed 2:  (a) Compute $\boldsymbol{F}_{B,L}$ using stored $\mathbf{M}_S^{-1}$.

           (b) Use stored $\mathbf{A}_{L,0}$ and $\tilde{\mathbf{A}}_{L,j}$.

**Appendix: Laplace Single Pert.**

    Seed 1:  (a) Compute $\boldsymbol{F}_{B,L}$ and store $\mathbf{M}_S^{-1}$.

           (b) Simplify symbolic $s$-domain Eq. (30) and compute time solution via inverse Laplace (Eqs. (31) and (32)).

Seed 2:  (a) Compute $\boldsymbol{F}_{B,L}$ using stored $\mathbf{M}_S^{-1}$.

           (b) Use stored time-domain solution (Eqs. (31) and (32)).

**Appendix E: Nomenclature**





$1f$     Blade first flapwise mode

$\alpha_l$     Airfoil angle of attack as blade variable

$\beta_l$     Airfoil twist angle as blade variable

$\boldsymbol{F}_B$     State-space forcing input vector

$\boldsymbol{F}_L$     Linear model forcing vector

$\boldsymbol{F}_T$     Time-domain model forcing vector

$\boldsymbol{F}_{B,L}$     Linear model state-space forcing vector

$\boldsymbol{F}_{B,T}$     Time-domain model state-space forcing vector

$\boldsymbol{q}$     State vector

$\boldsymbol{x}$     Structural degrees of freedom vector

$\delta\xi_5(t)$     Floater pitch angle virtual work

$\delta a_l(t)$     Blade displacement amplitude virtual work

$\delta u_l(r,t)$     Blade displacement virtual work

$\Delta V_{0,l,shear}$     Blade shear inflow velocity variation

$\Delta V_{0,l}$     Blade inflow velocity variation

$\Delta V_{0,turb}$     Rotor spatially coherent turbulence inflow velocity
variation

$\delta^n$     Perturbation of order $n$

$\epsilon_j$     Wave spectrum stochastic phase shift

$\gamma$     JONSWAP spectrum enhancement factor

$\hat{A}_l(r,t)$     Blade element acceleration tracked in $\hat{x}' - \hat{y}'$ coordi-
nate system

$\hat{D}_l(r,t)$     Blade element position tracked in $\hat{x}' - \hat{y}'$ coordinate
system

$\hat{V}_l(r,t)$     Blade element velocity tracked in $\hat{x}' - \hat{y}'$ coordinate
system

$\hat{x}_{Bot}$     Spar-buoy submerged underwater draft

$\mathbf{A}$     State-space system matrix

$\mathbf{A}_L$     Linear model system matrix

$\mathbf{A}_T$     Time-domain model system matrix

$\mathbf{C}_A$     Aerodynamic damping matrix

$\mathbf{C}_S$     Structural damping matrix

$\mathbf{K}_S$     Structural stiffness matrix

$\mathbf{M}_S$     Structural mass matrix

$\Omega$     Constant rotational speed

$\Omega_M$     Floater pitch moment excitation frequency

$\omega_{1f}$     Blade first flapwise mode natural frequency

$\omega_{\xi_5}$     Floater pitch natural frequency

$\omega_{wave}$     JONSWAP spectrum wave frequency

$\phi_l$     Airfoil inflow angle as blade variable

$\phi_{1f}$     Blade first flapwise mode shape

$\Psi_l(t)$     Blade azimuthal angular position

$\rho$     Air density

$\rho_{water}$     Water density

$\tau$     Dynamic stall time constant

$\tau_{hydro}$     Hydrodynamic moment

$\tau_{nom}$     Nominal dynamic stall time constant

$\xi_5$     Floater pitching angle

$a$     Axial induction factor normal to rotor plane

$a'$     Tangential induction factor





| | | | |
|---|---|---|---|
| | $a_l$ | Blade deflection amplitude | 945 |
| 920 | $A_M$ | Floater pitch moment amplitude | |
| | $A_{Spar}$ | Spar-buoy cylinder cross-sectional area | |
| | $A_{wave,j}$ | JONSWAP spectrum wave amplitude | |
| | $c$ | Airfoil chord length | |
| | $C_{L,inv}$ | Airfoil inviscid flow lift coefficient | 950 |
| 925 | $C_{L,l}$ | Airfoil dynamic lift coefficient as blade variable | |
| | $C_{L,stall}$ | Airfoil fully separated flow lift coefficient | |
| | $C_{L,static}$ | Airfoil static lift coefficient | |
| | $d$ | Blade reference radial position from the root | |
| | $D_l(r,t)$ | Blade element distance from the floater basis | 955 |
| 930 | $D_{Spar}$ | Spar-buoy diameter | |
| | $DOF$ | Degree of freedom | |
| | $EOM$ | Equation of motion | |
| | $F_{hydro}$ | Hydrodynamic force | |
| | $F_{l,aero}$ | Blade aerodynamic load | 960 |
| 935 | $f_{s,l}$ | Dynamic stall separation function as blade variable | |
| | $f_{s,static,l}$ | Dynamic stall static separation function as blade variable | |
| | $FFT$ | Fast Fourier Transform | |
| | $FOWT$ | Floating Offshore Wind Turbine | |
| 940 | $g$ | Gravitational constant | |
| | $GF_{a_l}$ | Generalized aerodynamic blade force | 965 |
| | $h$ | Water depth | |
| | $H_s$ | JONSWAP spectrum significant wave height | |
| | $iFFT$ | Inverse Fast Fourier Transform | 970 |

$k(r)$   Blade sectional stiffness

$k_{wave,j}$ Wave number

$l$   Blade index

$L_b$   Blade length

$L_l$   Blade lift force per unit length

$lin$   Linearized variable

$LM$   Linear model

$LTI$   Linear time-invariant

$M$   Combined mass of nacelle and hub

$m(r)$   Blade mass per unit length

$M_F$   Floater pitch moment

$M_{aero}$ Aerodynamic moment for floater pitch motion

$N_b$   Number of blades

$p_l(r,t)$ Blade rate of change of angular momentum

$PSD$   Power spectral density

$r$   Blade radial position from the root

$SDRE$ Standard deviation relative error

$st$   Steady value of variable

$t$   Time

$t_i$   Time step of index $i$

$T_p$   JONSWAP spectrum peak period

$TDM$ Time-domain model

$TI$   Turbulence intensity

$u_l(r,t)$ Blade deflection

$V_0(H)$ Rotor constant inflow at hub height

$V_{rel,l}$ Airfoil relative velocity as blade variable





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
