# Peer review of "Fast response methods for aero-elastic floating wind turbine design"

_Wind Energy Science, 2025_

## Referee Comment (RC1)

**Fast response methods for aero-elastic floating wind turbine design by Bogdan Pamfil et al. Review 1, 09.01.2026**

**General comments**

The article builds on earlier work on fast frequency-domain methods for analysis of floating wind turbines dynamics and adds details to the rotor blade aerodynamics by new methods to overcome the challenges with time-variant azimuthal dependencies in the system matrices. In my opinion, this is a significant step forward to supplement time-consuming time-domain methods with fast methods giving enough detail to support optimization studies during early-stage design.

The main scientific significance lies in the application of established and well documented mathematical methods to address fundamental challenges of rotor aerodynamics. So far, the focus of fast frequency-domain engineering methods for floating wind turbines have focused on the floater, and lumped the rotor loads to overall thrust and torque vectors. This article opens the path for comprehensive and fast floating wind turbine models with details of both rotor and floater.

The scientific quality is excellent and convincing. The derivations are detailed enough for other teams to reproduce and further build on the work. The step-by-step comparisons of the different models, converging towards identical results as more details are included, is a good indication of the quality of the work.

I enjoyed reading the article also due to the good presentation quality. Despite the large scope of the article, it is relatively easy to get an overview of the work and the main outcomes.

I think the article will have a large impact on future development of fast engineering models for floating wind turbines and should inspire other teams to include some of these ideas into their modeling efforts.

In my opinion, this article is ready for publication after adding a section on future work, and addressing some minor questions, clarifications and edits.

**Specific questions and comments**

As far as I can tell, hydrodynamic damping is not taken into account in this model. I understand that the focus here is on fast methods and rotor aerodynamics, but I nevertheless suggest some comments on how to incorporate more details of the floater in this modeling framework. Due to the simple structure of the structural damping matrix described in line 140, the pitch damping given by $Cs(1,1)$ can be scaled separately to take hydrodynamic damping into account. For large-volume floaters, results from Linear-

Potential-Theory (LPT) models such as WAMIT provide frequency-domain results for excitation, damping and added mass that can be incorporated as more floater degree-of-freedoms (DOFs) are added. Comments like this belongs perhaps under "future work".

Line 170 (L170): The way I read this section, Cl is evaluated only at 0.7R. I agree that for a well designed rotor, operating at optimum tip-speed ratio, this gives a good idea about the angle-of-attack and Cl for a large part of the outer part of the blade, and thereby the total thrust and torque. Please include (if I did not overlook it) a description on how the overall rotor loads are computed from the aerodynamics evaluation at 0.7R.

L420: "....and FB,L(ti) is assumed to be constant during that time interval." Is that an ad-hoc decision or are there numerical arguments for this choice? Some staggered methods for time stepping use the forcing at the midpoint of the step to achieve second-order accuracy in time. Has this been considered, is it relevant here?

L477: " .. a turbulence intensity (TI) of 5.77%." Would higher, but still realistic (say up to 15%) TI challenge modeling assumptions such as the linearizations?

Conclusions:

I miss a section on future work. Is it of interest to increase the level of detail, or will the linearizations nevertheless be the main source of error? What would be the next steps? Some examples, are they straight forward?

1. Evaluate aerodynamics at several radial stations.
2. More blade modes
3. Turbulence with variation across the rotor plane
4. More detailed hydrodynamics (LPT) of the floater, such as in QuLAF, and linearized floater drag terms.

Appendix A.

A1: Cl(alpha)?

It is shown in appendix A how the fully nonlinear force (A1)(which can be used directly in a time-domain model) is linearized and split into steady, forcing terms, damping terms and neglected higher order terms. But it is not totally clear to me what is used in the time-domain model. I suggest stating this explicitly here, even if it is mentioned earlier.

L790 ".... assuming additionally the same inflow angle $\phi_l$ for the entire blade.." The inflow angle usually vary significantly over the blade span, therefore structural twist optimization is important. The angle-of-attack varies to a lesser degree for a well-designed blade

operating at optimum tip-speed ratio. Please clarify. This is possibly connected to the comment on using one radial position to evaluate the forces for the complete rotor.

**Minor edits for consideration:**

L15: It is stated that the rotor loads are pre-computed. I think it would be clarifying to already here mention that aerodynamic damping due to the relative motion between structure and air is taken care of in the damping matrix. Distinguish between pre-computed aerodynamic excitation and aerodynamic damping computed during the simulation.

L18: I assume the damping is an outcome of the simulation, but this can also be read as the damping is pre-computed.

L66: The term "validated" has usually been used for comparison of model and experimental results or full-scale data. But lately I have seen it used also for comparison of models. Since this work is focused on mathematical methods, "validated" may be OK, but nevertheless consider if "verified" is more appropriate.

L84/85. Example of clarification: "In this paper, we expand the floating wind turbine model with wind loads for the blade elements of mass m(r), wave loads applied to the floater base, and gravity loads for the blade elements and hub and nacelle of cumulative mass M."

L94: "...that describes the rotational stiffness (component of the water plane stiffness matrix)"

L109: The term "rotating coordinate system" is often used for coordinate systems rotating with the rotor. Would "pitching coordinate system" be better?

L133: Although al has been defined earlier, it would be good to also here to state that a1, a2 and a3 are the blade deflection amplitudes for blade 1, 2 and 3.

L136: "The structural mass and damping matrices, as defined in Eqs. (11) and (13) of (Pamfil et al., 2025)..."

L157: "The hydrodynamic force is (pre-?) calculated with no spar motion consideration. Hydrodynamic damping is therefore not considered here, but can to some degree be taken into account by scaling the component (1,1) in the structural damping matrix Cs."

L223-227: The EOM of the TDM is combined with the original dynamic stall first-order ODE, and the LM's EOM is combined with the fully linearized dynamic stall model, right?

L344: Check eq. 21: qn(t), q0(t)?

L363:" As we supported with a more.... "  This sentence is heavy, consider rewriting

L 761: "The numerical methods that we elaborated are based on several assumptions and possess few limitations". Few limitations or a few limitations?

Check the nomenclature list for completeness. I did not check systematically but saw that Ns is missing.